# Temporal and compartment-specific signals coordinate mitotic exit with spindle position

Ayse Koca Caydasi[1,2,†], Anton Khmelinskii[3], Rafael Duenas-Sanchez[1,2], Bahtiyar Kurtulmus[1,2], Michael Knop[1,3] & Gislene Pereira[1,2]

The spatiotemporal control of mitotic exit is crucial for faithful chromosome segregation during mitosis. In budding yeast, the mitotic exit network (MEN) drives cells out of mitosis, whereas the spindle position checkpoint (SPOC) blocks MEN activity when the anaphase spindle is mispositioned. How the SPOC operates at a molecular level remains unclear. Here, we report novel insights into how mitotic signalling pathways orchestrate chromosome segregation in time and space. We establish that the key function of the central SPOC kinase, Kin4, is to counterbalance MEN activation by the cdc fourteen early anaphase release (FEAR) network in the mother cell compartment. Remarkably, Kin4 becomes dispensable for SPOC function in the absence of FEAR. Cells lacking both FEAR and Kin4 show that FEAR contributes to mitotic exit through regulation of the SPOC component Bfa1 and the MEN kinase Cdc15. Furthermore, we uncover controls that specifically promote mitotic exit in the daughter cell compartment.

[1] DKFZ-ZMBH Alliance, Department of Cell and Tumour Biology, German Cancer Research Centre (DKFZ), Im Neuenheimer Feld 280, 69120 Heidelberg, Germany. [2] Centre for Organismal Studies (COS), University of Heidelberg, Im Neuenheimer Feld 230, 69120 Heidelberg, Germany. [3] DKFZ-ZMBH Alliance, Centre for Molecular Biology (ZMBH), University of Heidelberg, Im Neuenheimer Feld 282, 69120 Heidelberg, Germany. † Present address: Department of Molecular Biology and Genetics, Gebze Technical University, 41400 Gebze/Kocaeli, Turkey. Correspondence and requests for materials should be addressed to G.P. (email: gislene.pereira@cos.uni-heidelberg.de).

Mitotic exit has been most extensively characterized in the model eukaryote *Saccharomyces cerevisiae*, yet a comprehensive understanding of how cells decide when to exit mitosis is still lacking. Biochemically, mitotic exit requires inactivation of the mitotic cyclin-dependent kinase (M-Cdk) and dephosphorylation of its targets. M-Cdk inactivation in budding yeast is under the control of a conserved proline-directed protein phosphatase, Cdc14. For the majority of the cell cycle, Cdc14 is retained in an inactivate state in the nucleolus through its association with the nucleolar protein Net1 (reviewed in ref. 1). In anaphase, Cdc14 is released from the nucleolus into the nucleus and cytoplasm in two waves. The first wave is triggered by the Cdc fourteen early anaphase release (FEAR) network, right after anaphase onset (reviewed in ref. 2). The FEAR network briefly releases Cdc14 from the nucleolus through the phosphorylation of Net1 by M-Cdk. This transient release is not sufficient to inactivate M-Cdk, but allows Cdc14 to dephosphorylate substrates important for anaphase functions, such as mitotic spindle stability, spindle elongation and rDNA segregation[2]. M-Cdk inactivation and full target dephosphorylation requires the second wave of Cdc14 release that is driven by the mitotic exit network (MEN)[3,4]. MEN is a signalling cascade localized at the spindle pole body (SPB, functional equivalent of the centrosome, reviewed in ref. 5). Therein, the small Ras like GTPase Tem1 triggers the MEN[4,6].

Faithful cell division is reliant upon the completion of two events before cells are permitted to exit mitosis: bipolar attachment of chromosomes to the mitotic spindle must occur before anaphase onset and sister chromatids must be segregated to the mother and daughter cells during anaphase. Two fail-safe mechanisms guarantee completion of these events: the spindle assembly checkpoint (SAC) and the spindle position checkpoint (SPOC). SAC (reviewed in ref. 7) prevents anaphase onset until all chromosomes are attached to the spindle microtubules in a bipolar manner (Fig.1a). SPOC, on the other hand, prevents MEN activation until the anaphase spindle is correctly aligned along the mother–daughter axis (Fig. 1a) (reviewed in ref. 8). The mother cell-localized kinase Kin4 is a key SPOC component[9–11]. When the anaphase spindle is misaligned, Kin4 keeps the Bfa1–Bub2 GTPase-activating protein (GAP) complex active. Bfa1–Bub2 in turn inhibits Tem1 to block MEN activation[12–17]. Without Kin4, the SPOC does not work.

Mitotic exit is tightly linked to the two checkpoints, SAC and SPOC, as their satisfaction is essential for FEAR network and MEN activation, respectively (Fig. 1a). In this way, the cell assures equal partitioning of the genomic DNA between the progenies before exiting from mitosis. FEAR network and MEN are also interlinked because even though FEAR network activity is not required for mitotic exit, it does enhance MEN activity (Fig. 1a)[18–20]. Finally, mitotic exit is also controlled by the inherent polarity of the budding yeast cell, although by less well understood mechanisms[21–27].

In this study we investigate how FEAR network, SPOC and polarity-associated factors communicate to assure faithful mitosis.

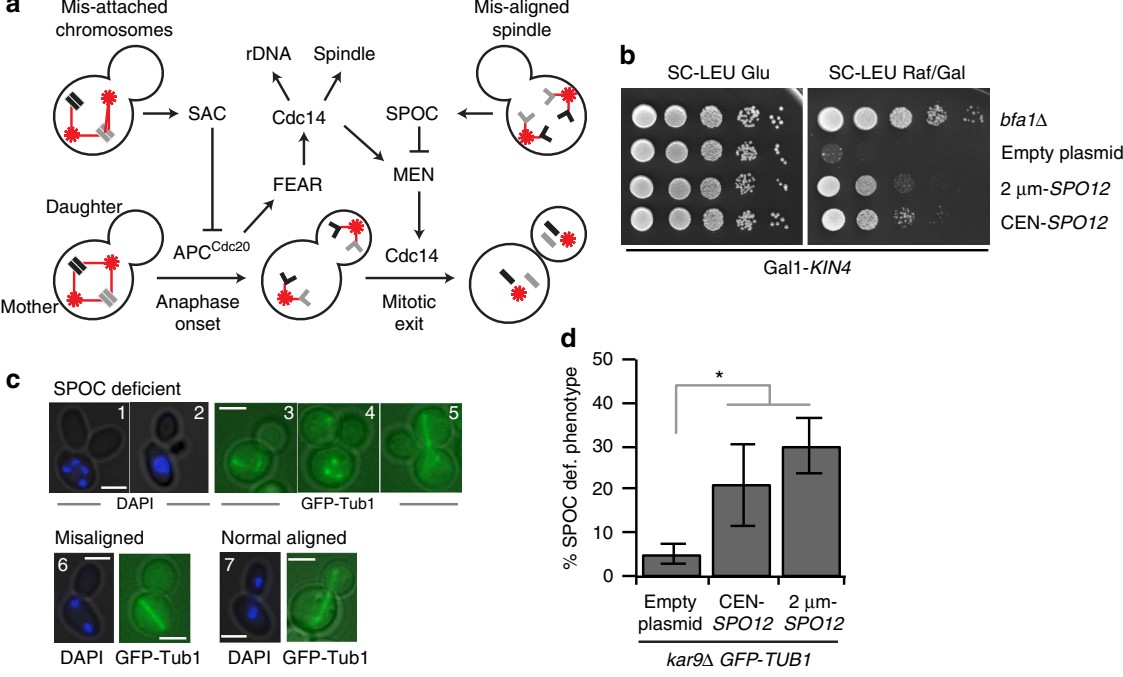

**Figure 1 | Spo12 promotes mitotic exit of cells with misaligned spindles. (a)** Cartoon depicting mitotic exit control in budding yeast. Arrowed and capped lines indicate activating and inhibitory steps, respectively. Spindle poles are depicted as red stars and microtubules as red lines. Chromosomes are shown in black and grey rectangles. Sister chromatids are outlined with light grey lines. **(b)** *SPO12* overexpression rescues *KIN4* overexpression lethality. Serial dilutions of Gal1-*KIN4* strains bearing high-copy (2 μm) or low-copy (centromeric (CEN)) plasmids carrying *SPO12*. Cells were spotted on Gal1-*KIN4* overexpressing (SC-LEU Raf/Gal) or suppressing (SC-LEU Glu) agar plates. Gal1-*KIN4* strain with *BFA1* deletion serves as a control for the rescue of *KIN4* overexpression lethality. **(c)** Examples of SPOC-deficient (1–5), misaligned nuclei/spindle (6) and correctly aligned nuclei/spindle (7) phenotypes. SPOC-deficient phenotypes arise from mitotic exit of cells with misaligned spindles. These include more than two nuclei in one cell body (1), multi-budded cells with clustered nuclei (2), broken spindle in one cell body (3, 4) and multi-polar spindle (5). Microtubules were monitored in cells carrying *GFP-TUB1* as a spindle marker. Nuclei were monitored by DAPI staining. Scale bars: 3 μm. **(d)** SPOC integrity of *kar9Δ GFP-TUB1* cells with or without additional copies of *SPO12* on a low-copy (CEN) or high-copy (2 μm) plasmid. To assay SPOC integrity, percentage of SPOC-deficient phenotypes per cell population were scored. Graph is an average of three independent experiments. Error bars show s.d. Per experiment, 100 cells were counted per strain. Asterisk indicates significant difference according to two-tailed Student's *t*-test (*P* < 0.05).

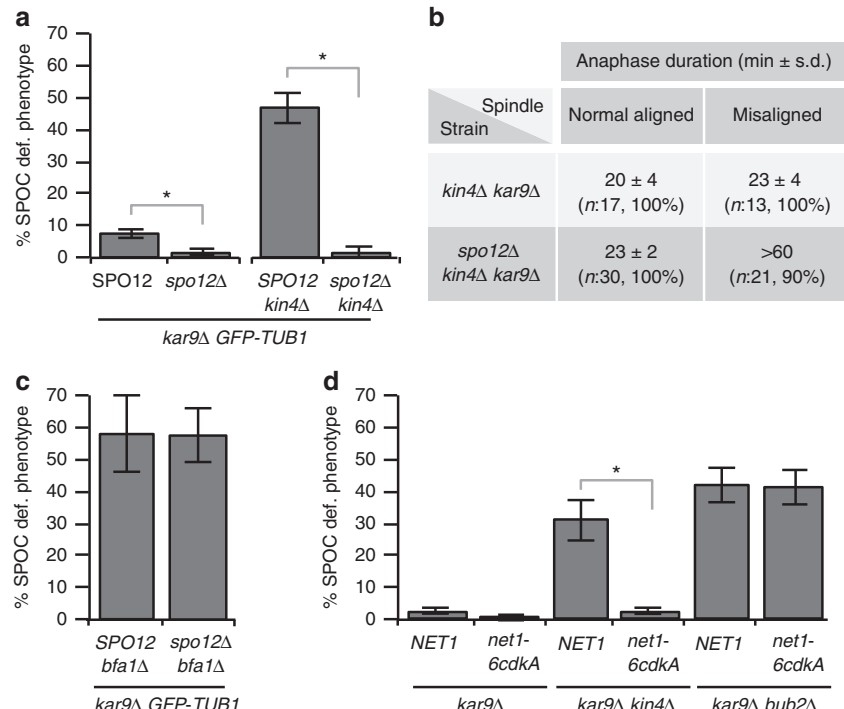

**Figure 2 | SPOC deficiency of *kin4Δ* cells is rescued by lack of FEAR.** (**a**) Percentage of SPOC-deficient phenotypes observed in *kar9Δ GFP-TUB1* cells with or without *KIN4* and/or *SPO12*. Graph is an average of five independent experiments. Error bars show s.d. Per experiment, 100 cells were counted per strain. Asterisk indicates significant difference according to two-tailed Student's *t*-test (*P* < 0.05). (**b**) Duration of anaphase in *kin4Δ kar9Δ* and *spo12Δ kin4Δ kar9Δ* cells during spindle normal alignment and misalignment established by time-lapse microscopy of corresponding *kar9Δ GFP-TUB1* cells. Duration of anaphase was calculated as the time elapsed from the start of fast spindle elongation (reference for metaphase-anaphase transition) until the spindle break down (reference for mitotic exit); *n*, number of cells analysed through time-lapse experiments with 1 min time resolution. Percentages show the prevalence of the indicated phenotypes. Note that *kin4Δ kar9Δ* cells exited mitosis within 20–23 min after metaphase–anaphase transition regardless of the spindle position. The anaphase took ~23 min when the mitotic spindle was normally aligned in *spo12Δ kin4Δ kar9Δ* cells. In addition, 90% of the *spo12Δ kin4Δ kar9Δ* cells with misaligned spindles stayed arrested with intact spindles for >60 min. (**c,d**) Percentage of SPOC-deficient phenotypes in indicated cells types. Graph is an average of three independent experiments. Error bars show s.d. Per experiment, 100 cells were counted per strain. Asterisk indicates significant difference according to two-tailed Student's *t*-test (*P* < 0.05).

Our data show that the main function of Kin4 is to counterbalance FEAR-dependent activation of MEN in the mother cell compartment. Our data also suggest that FEAR promotes mitotic exit through a dual mechanism of dephosphorylation of both Bfa1 and the MEN kinase Cdc15. Through a genome-wide genetic screen and by artificially targeting the daughter cell-associated proteins to the mother cell compartment, we further show that in the absence of FEAR, mitotic exit requires daughter cell-confined factors. These include the putative guanine nucleotide exchange factor (GEF) Lte1 and p21-activated protein kinase (PAK) Ste20. Our study therefore provides insight into how compartmentalized and timed events coordinately commit cells to exit mitosis.

## Results

**Elevated levels of Spo12 promote mitotic exit in the mother.** Overexpression of *KIN4* is lethal as it causes constitutive inactivation of the MEN GTPase Tem1 by the Bfa1–Bub2 GAP complex[9]. We identified *SPO12* as a multicopy suppressor of *KIN4* overexpression lethality (Fig. 1b). Inactivation of MEN components or overexpression of Bfa1 (refs 16,28) also invoked a late anaphase arrest. Overexpression of *SPO12* was able to suppress the lethality of *BFA1* overexpression (Supplementary Fig. 1a) and the temperature sensitivity phenotypes of *tem1-3*, *cdc15-1*, *dbf2-2*, *mob1-62* and *cdc5-10* but not *cdc14-2* MEN mutants[3] (Supplementary Fig. 1b,d). However, *SPO12* overexpression could not promote the growth of *tem1*, *cdc15*,

*dbf2/dbf20*, *cdc5* and *cdc14* null mutants (Supplementary Fig. 1c,d). Therefore, *SPO12* overexpression does not bypass MEN but promotes mitotic exit in a Cdc14-dependent manner.

We next asked whether overexpression of *SPO12* could promote mitotic exit in cells with misaligned anaphase spindles. To induce spindle misalignment, we used cells lacking the adenomatous polyposis coli-related spindle-positioning factor *KAR9* (ref. 29) (*kar9Δ*). In the absence of SPOC, *kar9Δ* cells with misaligned spindles exit mitosis and undergo cytokinesis despite their failure to segregate a copy of the genome into the daughter cell[14,30]. Therefore, SPOC-deficient *kar9Δ* cultures accumulate multinucleated cells (Fig. 1c[14,31]). Importantly, such multinucleated cells accumulated in *kar9Δ* cultures overexpressing *SPO12* (Fig. 1d, % SPOC-deficient phenotype) to indicate that high dosage of *SPO12* promotes mitotic exit regardless of the compartment in which the spindle elongates.

**Deletion of *SPO12* rescues SPOC deficiency of *kin4Δ* cells.** To understand whether endogenous *SPO12* influences SPOC function, we asked whether loss of *SPO12* influences mitotic exit in cells with misaligned spindles. Although the majority of *kar9Δ* cells with misaligned spindles are able to maintain the SPOC arrest, a small percentage of cells escape the arrest and become multinucleated[32,33] (Fig. 2a, *SPO12*, <10%). Remarkably, cells lacking *SPO12* (*spo12Δ*) accumulated fewer cells exhibiting the SPOC-deficient phenotype than *SPO12* cells (Fig. 2a). A more dramatic difference was observed in *kar9Δ* cells lacking *KIN4*.

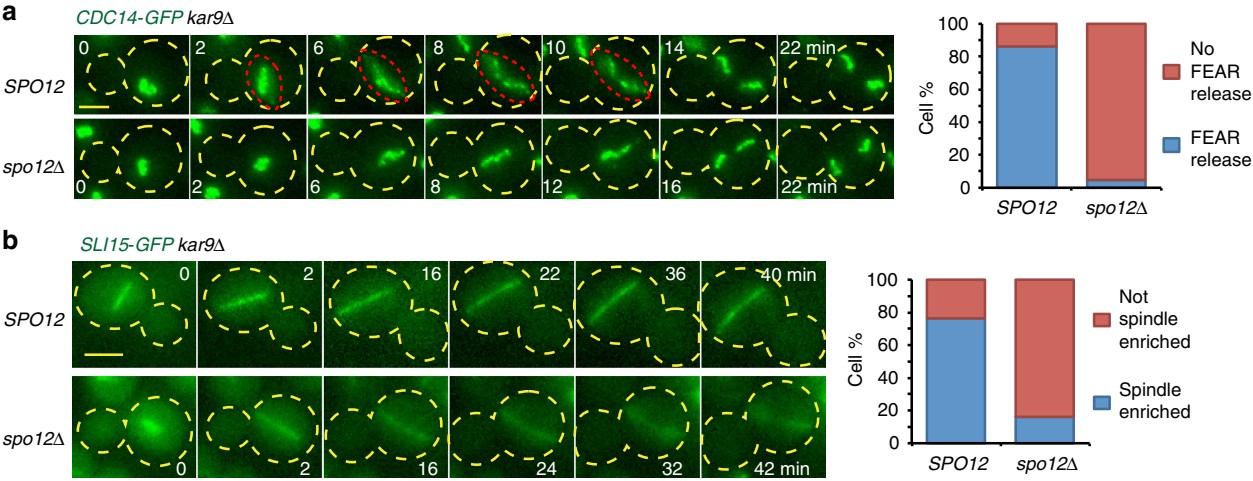

**Figure 3 | FEAR activity during spindle misalignment.** (**a**,**b**) Representative still images from the time-lapse series showing (**a**) Cdc14-GFP and (**b**) Sli15-GFP localization in the presence and absence of *SPO12*. The red dashed lines delineate Cdc14-GFP transiently released from nucleolus. The graphs show the percentage of cells with misaligned spindles with the indicated phenotypes. Per strain, 50 cells were analyzed by time lapse. Scale bars: 3 μm. The yellow dashed lines mark the cell boundary. Time point zero is the onset of anaphase (±1 min) based on the start of fast spindle elongation. Note that *spo12Δ* cells serve as a control where FEAR-released Cdc14 is blocked.

Surprisingly, deletion of *SPO12* completely rescued the severe SPOC deficiency of *kin4Δ* cells (Fig. 2a). This rescue was not due to a general block in cell-cycle progression, as deletion of *SPO12* in *kin4Δ kar9Δ* cells did not arrest the cells in anaphase when the mitotic spindle was correctly aligned (Fig. 2b). *SPO12* deletion also rescued the SPOC deficiency arising from the absence of other SPOC components in the Kin4 pathway (Supplementary Fig. 2a,b). However, deletion of *SPO12* did not suppress the SPOC deficiency of cells lacking *BFA1*, *BUB2* or carrying the GAP inactive *bub2^{R85A}* mutant[34] (Fig. 2c and Supplementary Fig. 2c,d). Together, these experiments suggest that *SPO12* promotes mitotic exit in cells with misaligned spindles and indicate that the Kin4 branch of the SPOC, but not Bfa1–Bub2 GAP activity, is dispensable for SPOC function in the absence of *SPO12*.

**Lack of FEAR prevents mitotic exit of *kin4Δ* in the mother.** Spo12 is part of the FEAR network that dissociates Cdc14 from its nucleolar inhibitor, Net1, through phosphorylation of Net1 by M-Cdk in early anaphase (Supplementary Fig. 3a)[2]. We sought to determine whether the control of mitotic exit of cells with misaligned spindles was specific for Spo12 or seen when FEAR network function was compromised. First, we analysed the *net1-6cdk* mutant, in which the six Cdk-phosphorylation sites of Net1 were mutated to alanine to prevent FEAR network-driven dissociation of Cdc14 from Net1 (ref. 35). Similar to *spo12Δ*, the *net1-6cdk* allele rescued SPOC deficiency of *kin4Δ* cells (Fig. 2d). Deletion of *SLK19*, another FEAR network component, also bypassed the requirement of *KIN4* for SPOC function (Supplementary Fig. 3b). These results indicate that Kin4 is not necessary for SPOC function in the absence of FEAR.

Cdc14 released by the FEAR network has multiple functions, including ribosomal DNA (rDNA) segregation, spindle midzone assembly and anaphase spindle elongation[2]. We therefore asked whether rDNA segregation fails during spindle misalignment in the absence of FEAR that might in turn invoke a SPOC-like mitotic arrest in *kin4Δ* cells. rDNA was segregated during spindle misalignment with the same timing as in cells with correctly aligned spindles (Supplementary Fig. 3c,d). Next, we asked whether deficiencies in spindle midzone or elongation functions would elicit a SPOC-like cell-cycle arrest in *kin4Δ* cells. Deletion

of *ASE1* or *CIN8*, which disrupts the spindle midzone integrity and elongation, respectively[36,37], did not rescue the SPOC deficiency of *kin4Δ* cells (Supplementary Fig. 3e). Thus, the ability of the removal of FEAR signalling to prevent mitotic exit of *kin4Δ* cells with misaligned spindles most likely occurs by a mechanism that is unrelated to the delay in rDNA segregation and spindle defects observed in FEARless cells.

**SPOC does not inhibit FEAR network-mediated Cdc14 release.** Given that the FEAR network contributes to unwanted mitotic exit in cells with misaligned spindles, we asked whether the FEAR network-mediated Cdc14 release and its downstream functions are inhibited in response to spindle misalignment. First, we analysed the release of *CDC14-GFP* using live-cell imaging. Cdc14-green fluorescent protein (GFP) was transiently released from the nucleolus into the nucleoplasm in cells with misaligned spindles after the metaphase–anaphase transition (Fig. 3a). This release occurred in a FEAR-dependent manner (Fig. 3a). Next, we analysed the localization of the chromosome passenger protein Sli15 to determine the activity of the transiently released Cdc14 during spindle misalignment. Sli15 must be dephosphorylated by Cdc14 released by the FEAR network to accumulate at the spindle during anaphase[38]. In cells with misaligned spindles, Sli15-GFP concentrated at the elongating anaphase spindle in a FEAR-dependent manner (Fig. 3b). Together, these results indicate that SPOC does not inhibit the transient FEAR network-mediated release of Cdc14 and that the SPOC arrest occurs despite Cdc14 activation in early anaphase.

**SPO12 deletion does not restore Kin4 regulation of Bfa1.** We sought to understand how the SPOC functions in *spo12Δ kin4Δ* cells. In wild-type cells, Kin4 regulates Bfa1 in two ways. First, Kin4 directly phosphorylates Bfa1 and, as a result, Bfa1 binds symmetrically (similar levels of Bfa1 at both SPBs) and dynamically to SPBs[14,26,30,39–41]. Second, Kin4 prevents phosphorylation of Bfa1 by Cdc5 to maintain the Bfa1–Bub2 GAP activity[10,11,42,43]. We therefore asked whether deletion of *SPO12* affects Bfa1 localization and Cdc5-dependent Bfa1 regulation in *kin4Δ* cells.

Through quantitative fluorescence microscopy and fluorescence recovery after photobleaching (FRAP) experiments we

found that deletion of *SPO12* had no impact on Bfa1 asymmetry and SPB-binding dynamics in *kin4Δ* cells with misaligned spindles (Supplementary Fig. 4). Next, we asked whether *SPO12* deletion prevented Cdc5 phosphorylation of Bfa1 in *kin4Δ* cells during spindle misorientation. Deletion of *KAR9* frequently generates cells with spindle misalignment, but this misaligned spindle is able to realign because of the Dynein-dependent spindle-positioning pathway[44]. To obtain a synchronous population of cells with persistent spindle misalignment, we generated a strain that allowed us to deplete Dyn1 (cytoplasmic dynein heavy chain) in *kar9Δ* cells[33]. The spindle midzone bundling protein *ASE1* was also deleted in these cells to prevent spindle reorientation because of spindle elongation. Once cells were released from G1 arrest under Dyn1-depleting conditions, the vast majority of these cells progressed into anaphase with misaligned spindles that were unable to realign (Fig. 4a,b). Using this strain, we analysed the Bfa1 phosphorylation profile in anaphase cells during spindle misalignment. Cdc5-phosphorylated Bfa1 can be detected in SDS–polyacrylamide gel electrophoresis (SDS–PAGE) gels as slower migrating (hyperphosphorylated) bands that become evident in the absence of *KIN4* (refs 9–11,21,45,46 (Fig. 4a,c,d, arrows)). During spindle misalignment, hyperphosphorylated Bfa1 forms appeared in *kin4Δ* cells independently of the presence of *SPO12* (Fig. 4a), but dependent on Cdc5 (Fig. 4c,d), suggesting that *SPO12* deletion does not prevent Cdc5 phosphorylation of Bfa1. We further tested this hypothesis in another experimental setup. In this setup, we forced Cdc5-GFP to constitutively associate with the SPB outer plaque component Spc42 by fusing Spc42 to the GFP-binding protein (GBP)[21] (GFP–GBP strategy). Constitutive targeting of Cdc5 to the SPB promoted hyperphosphorylation of Bfa1 and SPOC deficiency (Fig. 4e,f). Importantly, deletion of *SPO12* rescued the SPOC deficiency of Cdc5-GFP Spc42-GBP cells (Fig. 4f), but not the hyperphosphorylation of Bfa1 in these cells (Fig. 4e). These data strongly indicate that Cdc5 phosphorylates Bfa1 in cells lacking FEAR. Furthermore, *SPO12* deletion did not affect Cdc5 localization or ability of Cdc5 to phosphorylate other substrates at the SPB outer plaque in cells with misaligned spindles (Supplementary Fig. 5). Together, these data show that an absence of FEAR does not restore regulation of Bfa1 in *kin4Δ* cells to wild-type cells, but facilitates the SPOC arrest by alternative means.

**Cdc15 dephosphorylation at Cdk sites bypasses SPOC.** In cells with correctly aligned spindles, Cdc14 released by the FEAR network enhances MEN activity via dephosphorylation of Cdc15 and Mob1 (refs 18–20,47). Dephosphorylation of Cdc15 promotes its binding to SPBs[19,47] that then recruits more Dbf2-Mob1 (refs 48,49). Mob1-GFP signal intensity at SPBs is therefore a reflection of local Cdc15 activity. Following this rationale, we asked whether the mitotic arrest of *spo12Δ kin4Δ* cells with misaligned spindles is mediated by inhibition of the downstream MEN components Cdc15 and Mob1 by monitoring Mob1-GFP localization during spindle misalignment through live cell-imaging (Fig. 5a–d). In the presence of FEAR and Kin4, the levels of Mob1-GFP at SPBs remained low when the spindle was misaligned (Fig. 5a, cell 2) (for comparison, cell 1 enters anaphase with a correctly aligned spindle, Fig. 5a). Deletion of *KIN4* promoted SPB accumulation of Mob1-GFP (Fig. 5b,d). Mob1-GFP at SPBs of *kin4Δ* cells was under FEAR control as deletion of *SPO12* prevented this accumulation (Fig. 5c–e). Similar results were obtained for the SPB localization of Cdc15-GFP (Fig. 5e). Thus, lack of FEAR prevents the accumulation of Cdc15 and Mob1 at SPBs in *kin4Δ* cells with misaligned spindles. These data also suggest that Cdc14 released by the FEAR network dephosphorylates Cdc15 during SPOC

arrest, in agreement with our observation that faster migrating forms of Cdc15 appear after anaphase onset, regardless of the spindle position (Supplementary Fig. 6).

To understand the significance of Cdc15 and Mob1 regulation by FEAR for SPOC function, we analysed the checkpoint integrity in *cdc15-7A* and *mob1-2A* mutants that mimic Cdc15 and Mob1 that have been constitutively dephosphorylated by Cdc14, respectively[19]. We found that *cdc15-7A* (but not *mob1-2A*) cells accumulated SPOC-deficient phenotypes in both the presence and absence of FEAR (Fig. 5f). This suggests that constitutive Cdc15 dephosphorylation[18–20] bypasses the SPOC in the mother cell compartment. Nevertheless, fewer cells with SPOC-deficient phenotype accumulated in *kar9Δ cdc15-7A spo12Δ* cultures than those of *kar9Δ cdc15-7A* (Fig. 5f), indicating that Cdc14 released by the FEAR network might counteract mitotic arrest through additional mechanisms.

**Phosphorylation of Bfa1 at Cdk sites is required for SPOC.** Bfa1, a known substrate of Cdc14, could be another target of the FEAR network activated Cdc14 (ref. 50). The analysis of Bfa1 mobility shift during a MEN block (late anaphase arrest obtained by blocking MEN through Tem1 depletion) revealed that slow migrating forms of Bfa1-3HA accumulated in the absence (*spo12Δ*) but not in the presence (*SPO12*) of FEAR (Fig. 6a). This indicated that Cdc14 released by the FEAR network might promote Bfa1 dephosphorylation. Because Cdc14 dephosphorylates sites previously phosphorylated by Cdk[51,52], we asked whether Bfa1 is a substrate of M-Cdk. M-Cdk (Cdc28-as/Clb2) successfully phosphorylated bacterially purified MBP-Bfa1 *in vitro*, but not the MBP-Bfa1-6A in which the six Cdk consensus sites had been mutated to alanine (Fig. 6b and Supplementary Fig. 7a). Mass spectrometry analysis confirmed that M-Cdk phosphorylates Bfa1 *in vitro* at all six Cdk consensus sites (Supplementary Fig. 7b,c). Bfa1-6A was not phosphorylated by M-Cdk (Fig. 6b) but Bfa1-6A and Bfa1 were equally well phosphorylated by Cdc5 *in vitro* (Supplementary Fig. 8).

Next, we constructed yeast strains containing Cdk phophomimetic (S/T to D, *bfa1-6D*) or nonphosphorylatable (S/T to A, *bfa1-6A*) *BFA1* mutants at the *BFA1* endogenous locus. SPB localization of Bfa1-6A and Bfa1-6D resembled SPB localization of wild-type Bfa1 in cells with correctly and misaligned anaphase spindles (Supplementary Fig. 9a,b). This indicates that M-Cdk sites in Bfa1 do not affect recruitment of Bfa1 to SPBs. Cell-cycle progression of *bfa1-6A* and *bfa1-6D* cells were similar to *BFA1* wild-type cells in an unperturbed mitosis (Supplementary Fig. 9c). The protein levels of Bfa1-6A and Bfa1-6D were similar to Bfa1 (Supplementary Fig. 9d), yet the migration of the mutant proteins on SDS–PAGE gels slightly differed from that of wild-type Bfa1 in cells progressing normally through the cell cycle (Supplementary Fig. 9c). In MEN-blocked cells with normal aligned spindles, Bfa1-6D migrated slightly slower than the wild-type Bfa1 or Bfa1-6A on SDS–PAGE (Fig. 6c and Supplementary Fig. 9e). This behaviour of Bfa1-6D resembled Bfa1 in *spo12Δ* cells (Fig. 6c). Likewise, during spindle misalignment, Bfa1-6D migration was slightly slower than Bfa1 or Bfa1-6A, whereas phosphorylation profile of Bfa1-6A was similar to Bfa1 wild type (Supplementary Fig. 9f).

Both *bfa1* mutants retained functionality, as in contrast to *bfa1Δ* cells they were able to arrest upon microtubule depolymerization with the drug nocodazole (Fig. 6d). Interestingly, however, *bfa1-6A* (but not *bfa1-6D*) mutant was SPOC deficient (Fig. 6e). *bfa1-6A* also caused SPOC deficiency in *spo12Δ kin4Δ kar9Δ* but not in *spo12Δ kar9Δ* cells (Fig. 6e). These data suggest that in the absence of FEAR and Kin4, Bfa1 phosphorylation by M-Cdk is required to engage the SPOC arrest.

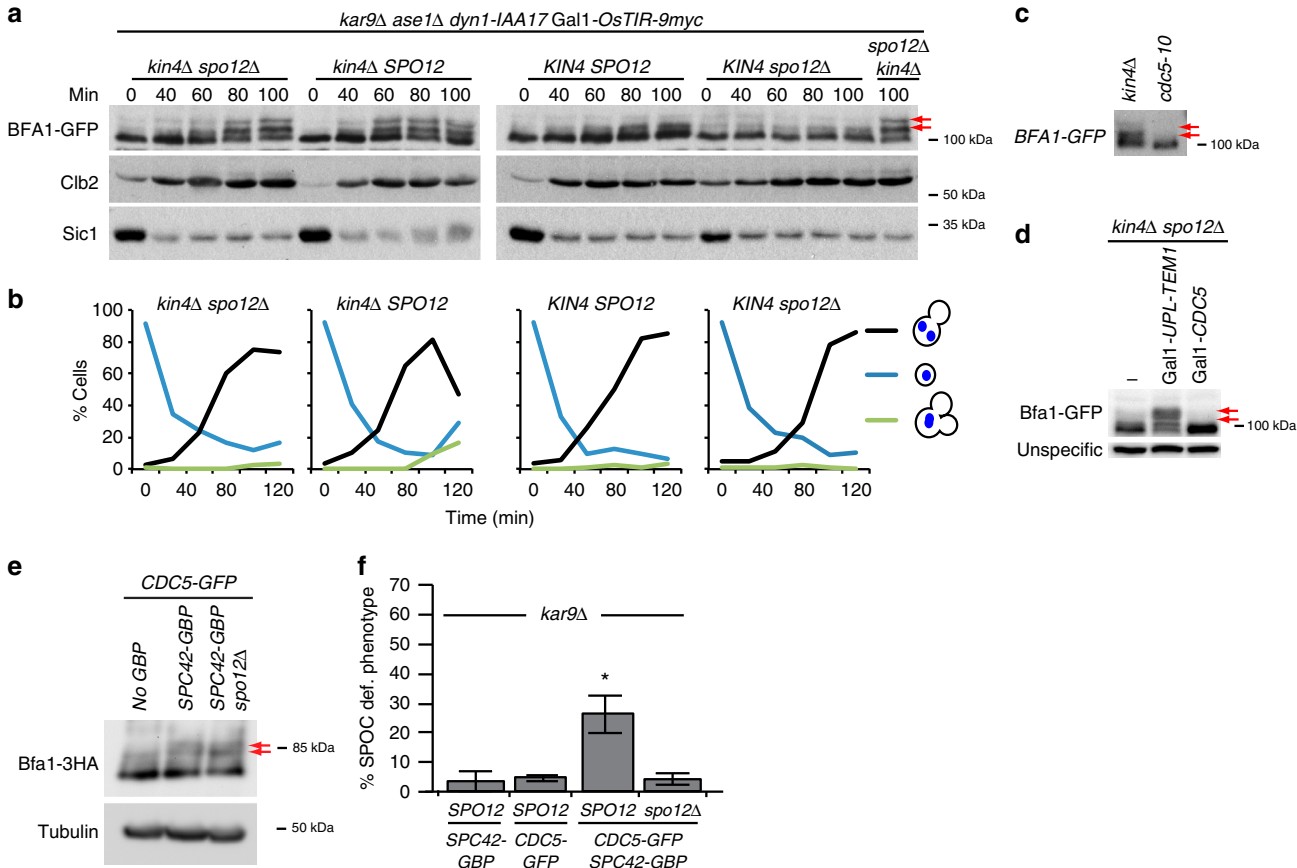

**Figure 4 | Bfa1 phosphorylation profile during spindle misalignment.** (**a,b**) Cells were released from G1 (time-point zero) arrest under *dyn1-IAA17* depleting conditions and samples were collected at indicated time points for immunoblotting and microscopy. (**a**) Immunoblots showing Bfa1-GFP mobility shift during spindle misalignment. Clb2 and Sic1 served as markers for cell-cycle progression. Arrows point to the hyperphosphorylated forms of Bfa1. (**b**) The graphs show the percentage of cells with misaligned nuclei per time point. Samples were analyzed by microscopy after DAPI staining. (**c,d**) Immunoblots showing Bfa1-GFP mobility shift during anaphase arrest, in temperature-sensitive *cdc5-10* cells at restrictive temperature (**c**) and in cells where Tem1 or Cdc5 were depleted (**d**). The 100 min sample of *kin4Δ* cells from the experiment shown in (**a**) was loaded as a reference for mobility shift in (**c**). Arrows point to the hyperphosphorylated forms of Bfa1. (**e,f**) Effect of artificial Cdc5 recruitment to the SPBs on Bfa1 phosphorylation profile and on SPOC integrity. (**e**) Immunoblot showing Bfa1-3HA mobility shift in logarithmic growing cultures of Cdc5-GFP cells, in the presence or absence of *SPC42-GBP* and *SPO12*. Arrows point to the hyperphosphorylated forms of Bfa1. (**f**) Percentage of SPOC-deficient phenotypes in indicated cells types. Graph is an average of three independent experiments. Error bars show s.d. Per experiment, 100 cells were counted per strain. Asterisk indicates significant difference of the indicated sample from other samples, according to two-tailed Student's *t*-test (*P* < 0.05).

If phosphorylation of Bfa1 by M-Cdk were required for Bfa1-Bub2 activity, one would expect *bfa1-6D* mutant to delay mitotic exit and impair growth. Mitotic exit was not delayed in Bfa1-6D cells (Supplementary Fig. 9c). However, the growth of *bfa1-6D* but not *bfa1-6A* cells was compromised upon mild Cdc5 inactivation (Fig. 6f) or Kin4 overproduction (Fig. 6g, Gal1-*KIN4*). No growth lethality was observed between Bfa1-6D and the temperature-sensitive MEN mutant *cdc15-1* (Supplementary Fig. 9g). This indicated that Bfa1-6D behaves as a more potent inhibitor of mitotic exit in conditions in which Bfa1–Bub2 GAP inactivation is compromised.

Together, our data indicate that Bfa1 (Fig. 6) and Cdc15 (Fig. 5) are key factors that contribute to the SPOC arrest of *kin4Δ spo12Δ* cells. In line with this interpretation, *bfa1-6A* and *cdc15-7A* mutants invoked SPOC deficiency in a synergistic manner (Fig. 6h). Moreover, unlike either single mutant alone, the SPOC deficiency of the combined *bfa1-6A cdc15-7A* mutant was not altered by *SPO12* deletion (Fig. 6h).

Mass spectrometry analysis revealed that T288, T340, S454 and T500 were phosphorylated in Bfa1 enriched from mitotic cells (Supplementary Fig. 10). Of these, T340 and T500 were not shown to be phosphorylated earlier[53–56]. We were unable to detect phosphorylation at S274 and T465, indicating that these sites may not be phosphorylated *in vivo* during mitosis[56] or phosphorylated at a level that is under the detection limit of our analysis. Of importance, a mutant form of Bfa1 carrying T288, T340, S454 and T500 substitutions to alanine (Bfa1-4A) was SPOC deficient similar to Bfa1-6A (Fig. 6i). This implied that these four Cdk phosphorylation sites detected *in vivo* are sufficient to promote Bfa1 function during SPOC arrest.

**Lte1 contributes to MEN activation independently of Kin4.** *kin4Δ spo12Δ* cells do not exit mitosis when the spindle is mis-aligned but exit mitosis when the anaphase spindle is correctly positioned (Fig. 2b). This reflects the presence of factors that specifically promote mitotic exit in *kin4Δ spo12Δ* cells when the spindle is correctly oriented along the mother–daughter cell axis. Lte1 is a bud-confined putative GEF, originally described as a potential GEF for Tem1 (refs 6,57,58) but recently shown to be an inhibitor of Kin4 in the bud[21–23]. Artificial targeting of Lte1 to the mother cell cortex causes mitotic exit in the mother cell

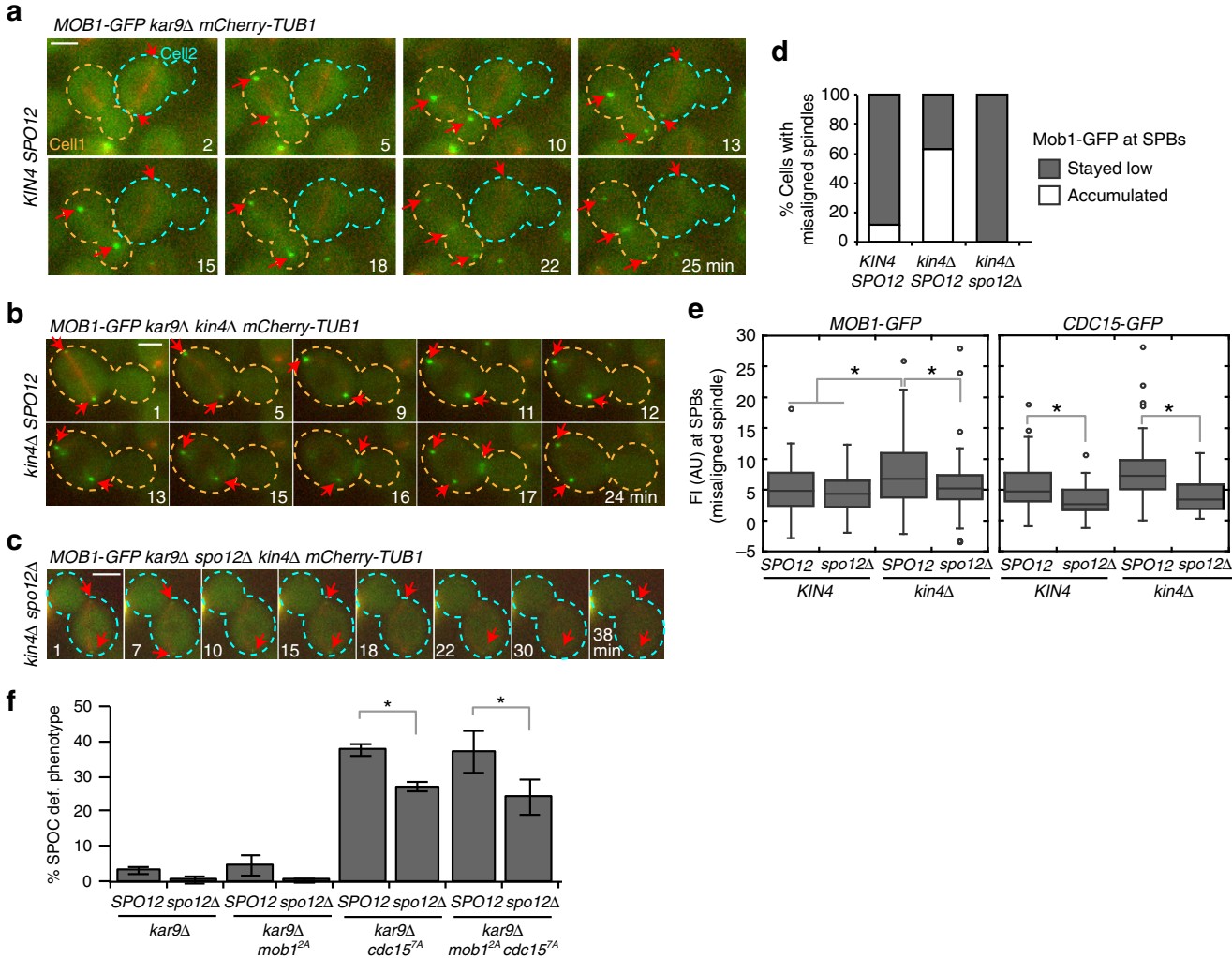

**Figure 5 | Mob1 and Cdc15 localization during spindle misalignment.** (**a–c**) Representative still images from the time-lapse series of *MOB1-GFP mCherry-TUB1 kar9Δ* cells with misaligned spindles in the presence of *KIN4* and *SPO12* (**a**), in the absence of *KIN4* (**b**) and in the absence of *KIN4* and *SPO12* (**c**). In (**a**), cell 1 has a normal aligned spindle, whereas cell 2 has a misaligned spindle. Arrows indicate the SPB or bud neck localized Mob1 when detectable. Scale bars: 3 μm. (**d**) Analysis of Mob1-GFP SPB binding during spindle misalignment. Time-lapse series of 30 cells were analyzed per strain. The graph shows the percentage of cells with misaligned spindles in which Mob1-GFP remained low or accumulated at SPBs during anaphase. (**e**) Box-and-whisker plots showing the mean fluorescence signal intensities of Mob1-GFP and Cdc15-GFP at SPBs of cells with misaligned spindles. The boxes show the lower and upper quartiles, the whiskers show the minimum and maximal values excluding outliers; outliers (shown as dots) were calculated as values greater or lower than 1.5 times the interquartile range; the line inside the box indicates the median. Asterisks mark significant differences according to two-tailed Student's *t*-test ($P < 0.01$). For Mob1-GFP plot, 100 SPBs were quantified per strain. For Cdc15-GFP plot, 50 signals were quantified per strain. (**f**) Percentage of SPOC-deficient phenotypes in the indicated cell types. Graphs show an average of three independent experiments. Error bars show s.d. Per experiment, 100 cells were counted per cell type. Asterisk indicates significant differences according to two-tailed Student's *t*-test ($P < 0.05$).

compartment (SPOC deficiency)[21], suggesting that Lte1 may be the mitotic exit-promoting factor in *kin4Δ spo12Δ* cells.

We next asked whether recruitment of Lte1 to the mother cell compartment would be able to drive *kin4Δ spo12Δ* cells with misaligned spindles out of mitosis. To address this question, we fused GBP to Sfk1, a plasma membrane protein localized specifically in the mother compartment[21]. Sfk1-GBP recruits Lte1-GFP to the mother cell cortex[21] (Fig. 7a). The mother-targeted Lte1 (*LTE1-GFP SFK1-GBP*) caused mitotic exit in *kin4Δ spo12Δ* cells with misaligned spindles (Fig. 7b). Thus, Lte1 has a Kin4-independent function in promoting mitotic exit. Analysis of Lte1 mutants[23] further suggested that this function requires the domain of Lte1 that was shown to bind to the GTPase Ras2[23,57], rather than Lte1 C-terminal GEF activity (Fig. 7c). Ras proteins could have a structural function in Lte1 activation. If this were the case, mother cell-targeted Lte1 would be unable to cause SPOC

deficiency in cells lacking Ras GTPases. This was not the case (Fig. 7d), implying that physical interaction of Lte1 with Ras proteins *per se* is not required for Lte1 function in mitotic exit.

Next, we postulated that if Lte1 is the only factor that activates mitotic exit in *kin4Δ spo12Δ* cells with appropriately aligned spindles, deletion of *LTE1* in this background should cause lethality. Cells lacking *LTE1 KIN4* and *SPO12* were however viable (Supplementary Fig. 11a)[9], indicating that Lte1 contributes to, but is not essential for, mitotic exit in *kin4Δ spo12Δ* cells.

**Ste20 promotes mitotic exit in the absence of FEAR and Lte1.** The viability of the *kin4Δ spo12Δ lte1Δ* mutant suggests that factors other than Lte1 promote mitotic exit. Similar to Lte1, these factors—once artificially targeted to the mother cell body—should promote mitotic exit in cells with misaligned

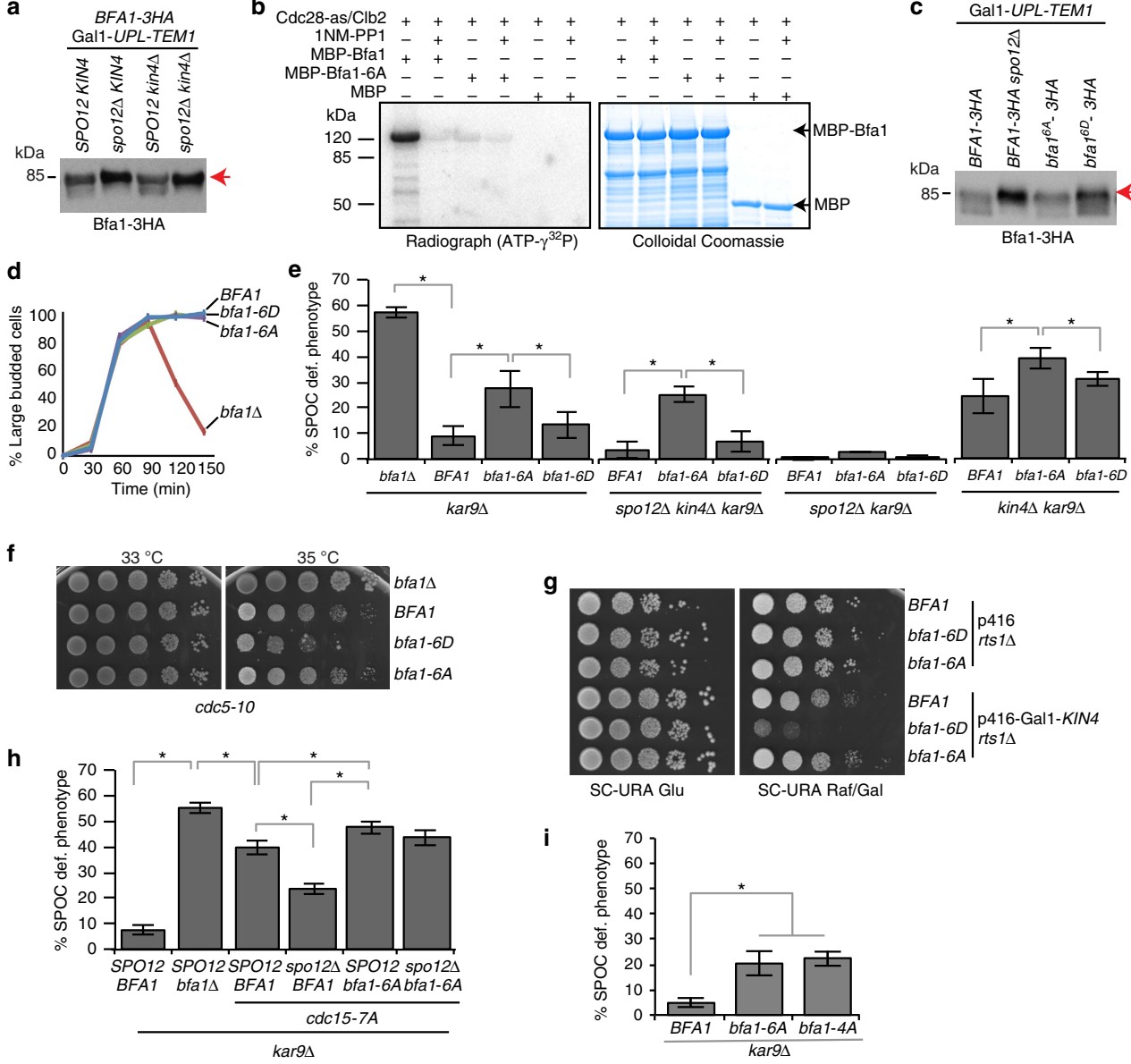

**Figure 6 | M-Cdk phosphorylates Bfa1.** (**a**) Bfa1-3HA migration profile on SDS–PAGE during late anaphase (Tem1 depletion). Arrow indicates the slow migrating form of Bfa1. Note the presence of faster migrating forms of Bfa1 in *SPO12*-containing strains (the first and the third lanes). (**b**) *In vitro* phosphorylation of Bfa1 fused to Maltose-binding-protein (MBP-Bfa1). The analogue-sensitive (as) mitotic M-Cdk complex (Cdc28-as/Clb2-TAP) purified from yeast was incubated with bacterially purified MBP-Bfa1, MBP-Bfa1-6A or MBP in the presence of DMSO or the ATP analogue 1NM-PP1. Incorporation of $\gamma^{32}$P-ATP into MBP-Bfa1 and the levels of MBP-Bfa1 are shown on the left and right, respectively. (**c**) Immunoblot showing migration profile of Bfa1 mutants. Cultures were released from G1-block under Tem1-depleting conditions. Arrow indicates the slow migrating form of Bfa1. Samples were collected 105 min after release (see supplementary Fig. 9e for the entire time course). (**d**) SAC proficiency of *bfa1Δ*, *BFA1*, *bfa1-6A* and *bfa1-6D* cells. Cells were released from G1-block ($t = 0$) in nocodazole containing medium. Nocodazole depolymerizes microtubules, provoking metaphase arrest. Note that all cell types except *bfa1Δ* accumulated as large budded mitotic cells, indicating cell-cycle arrest. Per strain, 100 cells were counted per time point. Graph is a representative of three independent experiments. (**e**) Percentage of SPOC-deficient phenotypes for the indicated strain backgrounds. Graphs show an average of three independent experiments. Error bars indicate s.d. Per cell type, 100 cells were counted. Asterisk indicates significant differences according to two-tailed Student's *t*-test ($P < 0.05$). (**f,g**) Growth analysis of Bfa1 mutants. (**f**) *cdc5-10 bfa1Δ* temperature-sensitive strains without or with integrated *BFA1*, *bfa1-6A*, *bfa1-6D* were grown at permissive (30 °C) and semipermissive (35 °C) temperatures. (**g**) *rts1Δ bfa1Δ* cells without or with integrated *BFA1*, *bfa1-6A*, *bfa1-6D* were spotted on glucose (glu) or galactose (Raf/Gal) containing agar plates. Cells contained either empty or Gal1-*KIN4* containing *URA3*-based 2 μm plasmid (p416). Note that deletion of *RTS1* rescues the lethality of *KIN4* overexpression[46,69]. (**h,i**) Percentage of SPOC-deficient phenotypes. Each bar is an average of three independent experiments. Error bars show s.d. Per experiment, 100 cells were counted per strain. Asterisks indicate difference according to two-tailed Student's *t*-test ($P < 0.05$).

spindles. To identify such factors, we performed a genome-wide screen to search for genes required for the growth of *spo12Δ lte1Δ kin4Δ* but not *spo12Δ kin4Δ* or *lte1Δ kin4Δ* mutants. Using synthetic genetic array technology[59], we crossed each mutant with a collection of strains carrying individual deletions of nonessential genes and measured the size of the resulting colonies (Supplementary Fig. 11b). This analysis identified 43 gene deletions that specifically impaired growth of the *spo12Δ lte1Δ*

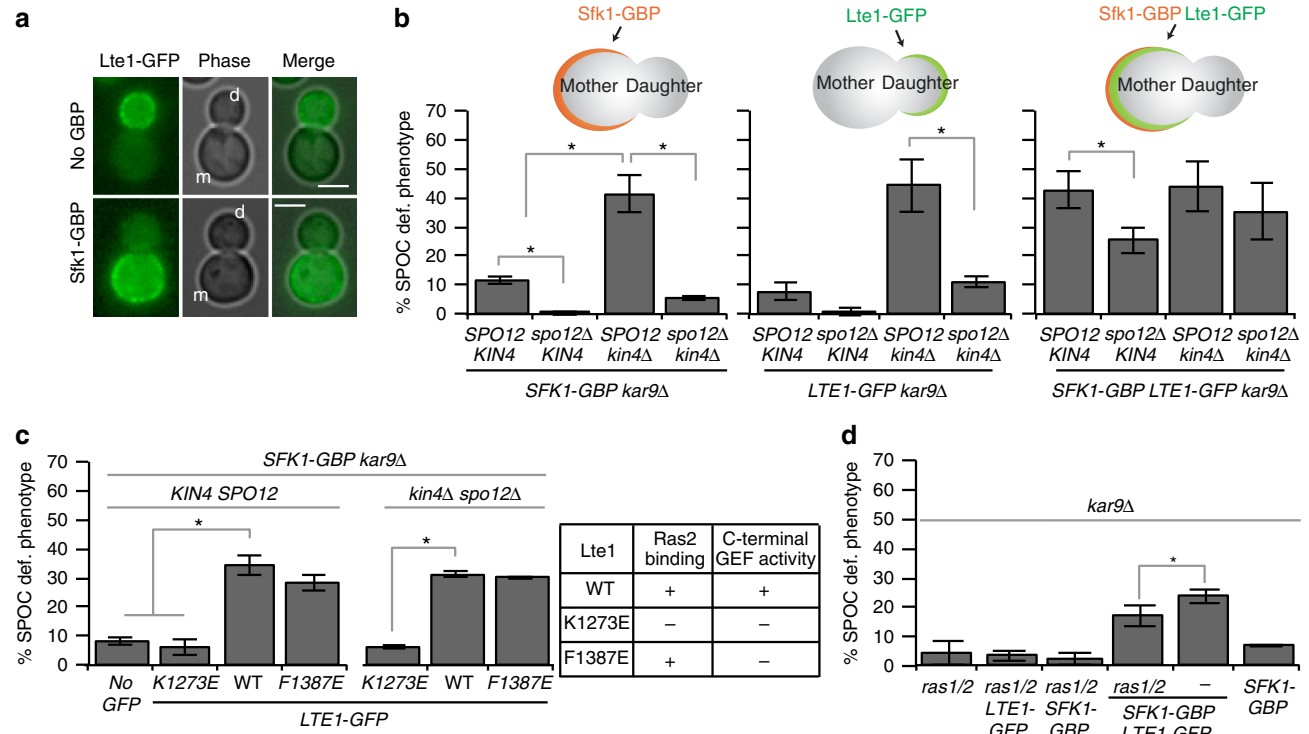

**Figure 7 | Lte1 has a Kin4-independent function in mitotic exit.** (**a,b**) Targeting of Lte1-GFP to the mother cell cortex via Sfk1-GBP affects SPOC integrity. (**a**) Representative images showing Lte1-GFP localization in the absence or in the presence of Sfk1-GBP; m, mother, d, daughter cell. Scale bars: 3 µm. (**b**) Percentage of SPOC-deficient phenotypes observed in the indicated strain backgrounds. Other graphs serve as controls using the strains bearing *SFK1-GBP* (on the left) and *LTE1-GFP* (in the middle) separately. Cartoons depict the strain type and protein localization on each set of strains. Each graph is an average of three independent experiments. Error bars are s.d. Per experiment, 100 cells were counted per cell type. (**c**) K1378E mutation in *LTE1-GFP* abolishes SPOC deficiency when recruited to the mother cell. Two mutant forms of Lte1-GFP was targeted to the mother cell by Sfk1-GBP. Both K1273E and F1387E are GEF-inactivating mutations at Lte1 C-terminal domain based on homology to Cdc25 (ref. 70). K1273E also impairs Lte1 binding to Ras2 (ref. 70). The table on the right summarizes these properties. Each graph is an average of three independent experiments. Error bars show s.d. Per experiment, 100 cells were counted per cell type. (**d**) Percentage of SPOC-deficient phenotypes in indicated cell types. *ras1/2* corresponds to *ras1Δ ras2Δ* cells kept alive with overexpression of *TPK1*. Asterisk indicates significant difference according to two-tailed Student's *t*-test (*P* < 0.05). Each bar is an average of three independent experiments. Error bars show s.d. Per experiment, 100 cells were counted per cell type.

*kin4Δ* triple mutant (Supplementary Fig. 11c,d and Supplementary Data 1). Among the hits, 12 genes were functionally related to polarized cell growth, cytoskeletal organization and signalling (Supplementary Fig. 11e) that we considered to be candidate genes that could be involved in bud-specific activation of mitotic exit. Therefore, we analysed their ability to promote mitotic exit when targeted to the mother cell compartment using the GFP–GBP strategy[21]. Strikingly, artificial targeting of Ste20-GFP to the mother cell promoted mitotic exit of *spo12Δ kin4Δ* and *SPO12 KIN4* cells with misaligned spindles (Fig. 8a–c). Therefore, Ste20 plays a critical role alongside Lte1 in promoting mitotic exit in cells lacking FEAR and Kin4. However, because of the function of Lte1 as a Kin4 inhibitor, *LTE1* but not *STE20* becomes essential in the absence of FEAR (Fig. 8d). Previously, mother cell-enriched Lte1 was shown to promote mitotic exit in cells with misaligned spindles but not in cells treated with nocodazole that arrest in metaphase because of SAC activation[21]. Similarly, mother cell-enriched Ste20 promoted SPOC but not SAC deficiency (Supplementary Fig. 11f). This implies that, like Lte1, Ste20 cannot promote mitotic exit before the metaphase-to-anaphase transition that is inhibited by SAC.

Ste20 is a p21-activated kinase that has been previously implicated in mitotic exit controls through a yet to be deciphered mechanism[24]. Ste20 has established functions in several mitogen-activated protein kinase (MAPK) signalling pathways, such as pheromone and osmosensory signalling, filamentous and pseudohyphal growth[60]. We found that Ste20 kinase activity drives mitotic exit in *spo12Δ kin4Δ lte1Δ* cells, as a catalytically inactivated (kinase-dead) version of Ste20 (ref. 61) was synthetically lethal with this genotype (Fig. 8e). We next asked whether the mitotic exit that is driven by Ste20 exploits the MAPK signalling pathways. Unlike the deletion of *STE20*, neither deletion of *STE11* (the downstream target of Ste20 in MAPK signalling[62]) nor *FUS3*, *KSS1* or *HOG1* (the eventual MAPKs activated by their corresponding MAPK signalling[62]) suppressed the growth of *kin4Δ spo12Δ lte1Δ* cells (Fig. 8f). Similarly, SPOC deficiency invoked by recruitment of Ste20 to the mother cell compartment did not require Ste11, Fus3, Kss1 or Hog1 (Fig. 8g). Thus, Ste20 does not work via the MAPK signalling in promoting mitotic exit. Importantly, *STE20* deletion impaired growth of *spo12Δ lte1Δ kin4Δ* but not *spo12Δ lte1Δ bfa1Δ* cells (Fig. 8h), suggesting a role for Ste20 upstream of the Bfa1–Bub2 GAP.

## Discussion

We envisage a model where two determinative hubs promote mitotic exit: one centred on FEAR and the other operating at the level of Lte1 and Ste20. Lte1 and Ste20 constitute a spatial activator, ensuring mitotic exit when one of the two spindle poles enters the bud. The FEAR, on the other hand, is a timer that primes mitotic exit only after sister chromatid separation. The cell relies upon the activation of at least one of these two hubs to

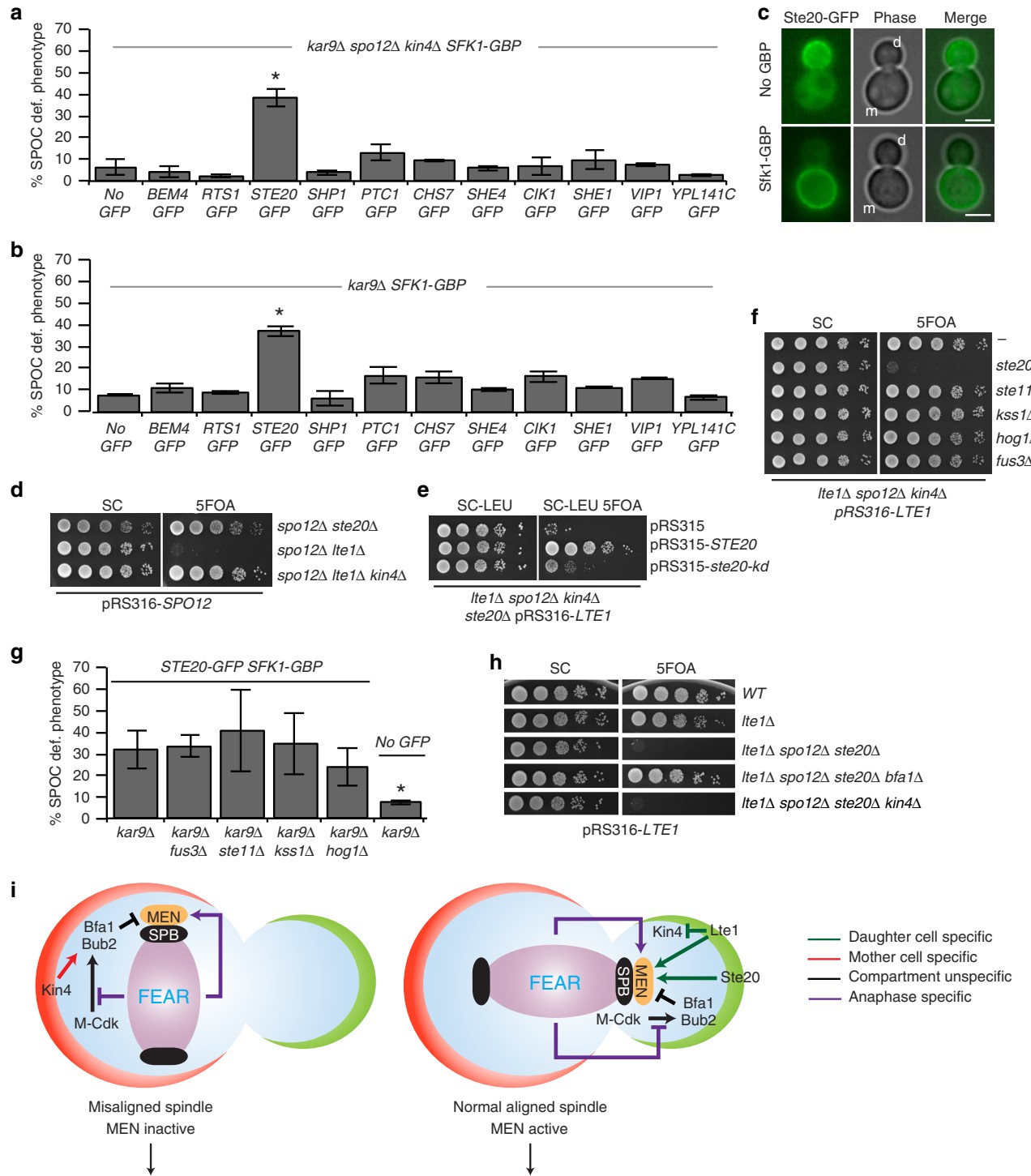

**Figure 8 | Ste20 promotes mitotic exit parallel to Lte1.** (**a**,**b**) SPOC integrity of *spo12Δ kin4Δ* cells upon mother cell targeting of the indicated GFP-tagged proteins through *SFK1-GBP*. Graph is an average of three independent experiments. Error bars show s.d. Per experiment, 100 cells were counted per cell type. Asterisk indicates significant difference from no-GFP control (two-tailed Student's *t*-test, $P < 0.05$). (**c**) Representative images showing Ste20-GFP localization in the absence (upper row) and presence (lower row) of Sfk1-GBP; m, mother, d, daughter cell. Scale bars: 3 μm. (**d**–**f**) Growth analysis showing serial dilutions of the indicated cell types. *SPO12* or *LTE1* deletion was complemented by the corresponding wild-type gene present on an *URA3*-based centromeric plasmid (pRS316). *URA3*-based plasmid loss was induced on 5FOA plates. Leucine lacking medium (SC-LEU) selects for *LEU2* based-centromeric plasmids (pRS315). *ste20-kd*: kinase inactive version of *STE20* (in **d**). (**g**) Percentage of SPOC-deficient phenotypes in *STE20-GFP SFK1-GBP* cells with indicated gene deletions. Graph is an average of three independent experiments. Per experiment, 100 cells were counted per cell type. Error bars show s.d. Asterisk indicates significant difference from *kar9Δ STE20-GFP SFK1-GBP* cells (two-tailed Student's *t*-test, $P < 0.05$). (**h**) Serial dilutions of indicated cell types bearing pRS316-*LTE1* (*LTE1* in *URA3*-based centromeric plasmid) were spotted on SC (nonselective) and 5FOA (negative selection for *URA3*-based plasmids). (**i**) Current model of MEN activation by compartment-specific and unspecific factors. Arrows indicate activation, whereas capped lines indicate inhibition.

promote mitotic exit (Fig. 8i). Because FEAR promotes MEN regardless of the specific compartment in which the nuclei resides, cells require the mother cell-specific kinase Kin4 to prevent mitotic exit when the anaphase spindle remains within the mother cell body (Fig. 8i).

The function of Kin4 in SPOC is to loosen the Bfa1–SPB association in order to isolate the GAP from its inhibitor, Cdc5 (refs 32,41,45). Strikingly, in the absence of FEAR activation of the MEN, the central SPOC kinase Kin4 was no longer required for SPOC. This also explains why deletion of *SPO12* rescues the SPOC deficiency of a Bfa1 version that is artificially anchored to SPBs[63]. Although Kin4 is essential to maintain Bfa1–Bub2 GAP activity, how does the SPOC function without Kin4 in the absence of FEAR? Our data show that SPOC relies on the activity of the Bfa1–Bub2 GAP complex in the absence of FEAR. The analysis of Bfa1 migration profile in SDS–PAGE gels indicated that the inhibitory phosphorylation of Bfa1 by Cdc5 still occurs in the absence of *SPO12* in *kin4Δ* cells or in cells where Cdc5 phosphorylation of Bfa1 was enhanced by artificial tethering of Cdc5 to the SPB. Yet, Cdc5 phosphorylated Bfa1 was able to maintain the SPOC arrest in the absence of FEAR. These data imply that the *in vivo* GAP activity of Cdc5-phosphorylated Bfa1–Bub2 is probably maintained at a level high enough to provide Tem1 inhibition in the mother cell, in the absence of FEAR and Kin4. In support of this notion, in *kin4Δ* cells Bfa1 is able to maintain the metaphase arrest in response to microtubule depolymerization even when Bfa1 is phosphorylated by Cdc5 (refs 9–11,16). In addition, Cdc5 was shown to only partially inactivate the Bfa1–Bub2 GAP complex *in vitro*[42].

We propose that in contrast to Cdc5, which works as an inhibitor of the GAP complex, Bfa1–Bub2 is positively regulated by Bfa1 phosphorylation at Cdk sites. Unlike many cases where Cdk phosphorylation of the substrate primes for Polo kinase phosphorylation of the same substrate[64], our data indicate that phosphorylation of Bfa1 Cdk sites does not interfere with the ability of Cdc5 to phosphorylate Bfa1 *in vitro* and *in vivo*. In fact, if Cdk phosphorylation of Bfa1 were to inhibit the ability of Cdc5 to downregulate GAP activity, Bfa1 non-phosphorylatable mutants would have been proficient in SPOC. Instead, we observed that Cdk non-phosphorylatable mutant (Bfa1-6A) imparted SPOC deficiency upon cells lacking both FEAR and Kin4. We therefore favour a model in which Kin4 and Cdk-dependent phosphorylation of Bfa1 work in parallel to keep the Bfa1–Bub2 GAP active (Fig. 8i). In support of this model, the accumulation of SPOC-deficient phenotypes increased upon inactivation of the two Bfa1 activating pathways, Kin4 and Cdk (*kin4Δ bfa1-6A* cells), both in the absence and presence of FEAR (Fig. 6e). Yet, we cannot exclude the possibility that Bfa1 dephosphorylation by FEAR-released Cdc14 might be prevented during spindle misalignment to engage the SPOC by an unknown mechanism independent of Kin4.

How does FEAR network contribute to MEN activation in *kin4Δ* cells with misaligned spindles? FEAR network activation promoted a transient release of Cdc14 out of the nucleolus during spindle misalignment. Cdc14 was shown to counteract Cdk phosphorylation of the MEN components Cdc15 and Mob1 (refs 18–20,65). Dephosphorylation of Cdc15 by Cdc14, in addition to Tem1 activation, is critical for the recruitment of Cdc15 to SPBs[19,48,66], where Cdc15 activates the Dbf2-Mob1 kinase complex[48,67]. We found that FEAR network promoted dephosphorylation of Cdc15 and SPB recruitment of Cdc15 and Mob1 in *kin4Δ* cells with misaligned spindles. Impairment of FEAR in *kin4Δ* cells lowered the levels of SPB-associated Cdc15 and Dbf2-Mob1, most likely explaining why the SPOC became functional in those cells. The SPB recruitment of Cdc15 seems to be a critical element that promotes mitotic exit in the mother cell

compartment. Consistently, a Cdc15 mutant that constitutively mimicked the dephosphorylated state of this protein bypassed the requirement of FEAR network to promote mitotic exit in the mother cell compartment. In contrast, Bfa1-6A only caused SPOC deficiency in FEARless cells when Kin4 was absent. This indicates that in comparison with Bfa1-6A, Cdc15-7A is a more potent MEN activator, and this is in line with the ability of Cdc15-7A to rescue the temperature sensitivity of MEN mutants *tem1-3* and *dbf2-2* (ref. 18). Yet, Tem1 inhibition by active Bfa1–Bub2 GAP complexes may explain why the FEAR-dephosphorylated Cdc15 is unable to trigger mitotic exit during SPOC; we cannot exclude that Kin4, or other SPOC components, may work downstream of Bfa1–Bub2 to provide a more robust cell cycle arrest.

Our genetic data indicate that the FEAR network and bud-confined factors are required for mitotic exit. Cells lacking Kin4 and FEAR network were unable to exit mitosis in the mother cell compartment unless the bud-localized, putative GEF Lte1 or the PAK-kinase Ste20 was recruited to this cellular compartment. Furthermore, cells with normal aligned spindles lacking *KIN4*, *SPO12*, *LTE1* and *STE20* were synthetic lethal and only able to divide when *BFA1* was deleted. We established that Lte1 promotes mitotic exit by a dual mechanism, one mode depends on Kin4, as reported previously[21,22], whereas the other acts through a Kin4-independent pathway (Fig. 8i). The Kin4-independent function of Lte1 most likely relies upon its Ras2 binding GEF domain (Ras GEF domain). This is in line with the observation that the growth defects of *lte1Δ* cells could not be rescued by Lte1 mutant proteins that cannot interact with the GTPase Ras2[23,57,68]. Ras proteins were shown to act upstream of Lte1 in promoting Lte1 binding to the bud cortex[57]. However, we found that *RAS* gene deletions did not impair the ability of mother cell-targeted Lte1 to induce mitotic exit. This implies that binding of Lte1 to Ras2 *per se* is not absolutely required for the mitotic-exit promoting function of Lte1, as long as Lte1 is targeted to the cell cortex by other means. How the Lte1 Ras GEF domain influences mitotic exit remains to be established. Lte1 Ras GEF domain could activate Tem1, as suggested by genetic analysis[6], or be required to inactivate the MEN inhibitors Kel1/Kel2 (ref. 24).

Through a genome-wide screen, we identified several genes that became essential in the absence of FEAR, Kin4 and Lte1, including *STE20*. When bud-confined Ste20 was drawn artificially into the mother cell, it was able to promote mitotic exit of cells with misaligned spindles. This suggests that, like Lte1, Ste20-dependent activation of mitotic exit is a daughter cell-specific phenomenon (Fig. 8i). However, neither Lte1 nor Ste20 could promote mitotic exit when recruited to the mother in cells treated with nocodazole, suggesting that events at the metaphase-to-anaphase transition other than FEAR, for example, anaphase-promoting complex activation, may be required for both Lte1 and Ste20 to exert their function in mitotic exit.

Previous genetic data identified *STE20* as a mitotic exit activator that acts redundantly alongside *LTE1* (ref. 24). However, deletion of *LTE1* is synthetic lethal with FEAR mutants[20] (that is, with *spo12Δ*), whereas the deletion of *STE20* is not. Our data now show that this distinction arises from the additional function of Lte1 in Kin4 inhibition, as the deletion of *KIN4* rescued *spo12Δ lte1Δ* lethality. We further show that the function of *STE20* in promoting mitotic exit is independent of the known function of Ste20 in MAPK signalling. How Ste20 activates MEN remains to be established. The fact that deletion of *BFA1* (but not *KIN4*) rescues the lethality of *ste20Δ lte1Δ spo12Δ* cells suggests that Ste20 most likely inactivates the Bfa1–Bub2 GAP complex or activates the MEN downstream of the inhibitory GAP, for example, at the level of the GTPase Tem1.

Our study helps to clarify how FEAR, MEN, SPOC and daughter cell-associated factors regulate mitotic exit in space and time through an intricate combination of independent pathways that apply control at the level of compartmentalization (Lte1, Ste20) and timing (FEAR). Whether the cell requires an intelligent sensor to detect spindle orientation defects, or solely relies on compartmentalization for the SPOC function remains to be clarified. The fact that Kin4 requires its recruitment to SPBs to arrest exit upon microtubule insult[9–11,33], however, is highly suggestive of presence of a sensory mechanism that would operate alongside the compartmentalized regulation of mitotic exit.

## Methods

**Synchronous induction of spindle misalignment.** To induce temporary spindle misalignment, log-phase kar9Δ cultures grown at 23 °C were shifted to 30 °C. For induction of a permanent spindle misalignment, kar9Δ ase1Δ DYN1-IAA17 Gal-OsTIR1-9myc cells were grown at 23 °C in YP-Raf/Gal medium to obtain a log-phase culture that was then shifted to 30 °C for ~2 h followed by α-factor addition. The culture was incubated for another ~3 h at 30 °C until >90% of the cells formed mating projections. Cells were then washed twice with fresh YP-Raf/Gal media and resuspended in prewarmed (30 °C) YP-Raf/Gal medium supplemented with 1.5 mM 3-Indoleacetic acid (Sigma). Culture was incubated at 30 °C and samples were collected every 15–20 min for ~2–3 h.

**SPOC integrity assays.** For population analysis of SPOC integrity, log-phase kar9Δ cells cultured at 23 °C were shifted to 30 °C and maintained at 30 °C for 3–5 h. Cells were fixed using 70% ethanol and resuspended in phosphate-buffered saline (PBS) containing 1 μg ml$^{-1}$ 4′,6-diamino-2-phenylindole (DAPI, Sigma). Images were taken using differential interference contrast and ultraviolet illumination to reveal DAPI staining. Cells with normal and misaligned nuclei, and cells with SPOC-deficient phenotypes, such as multiple nuclei in one cell body, or single nuclei in a multi-budded cell, were scored. kar9Δ cells bearing GFP-TUB1 were fixed with 4% paraformaldehyde for 10 min at room temperature before image acquisition in differential interference contrast and GFP channels. Cells with normal and misaligned spindles and cells with SPOC-deficient phenotypes, such as broken spindles, or multiple SPBs in one cell body, or multipolar spindles, were scored. A total of 100 cells were counted per strain per experiment. Sample sizes were determined a priori with an anticipated effect size of 0.4, statistical power level of 0.8 and a probability level of 0.05 for a two-tailed t-test. The summation of the SPOC-deficient phenotypes was plotted as a percentage of total cells. Experiments were repeated independently 3–5 times (the number of repeats are indicated in the figure legend) and the average SPOC deficiencies and s.d. were calculated, and plotted. SPOC deficiencies from 3 to 5 independent experiments were compared using two-tailed Student's t-test. For analysis of SPOC integrity through time-lapse movies, GFP-TUB1 kar9Δ cells grown at 23 °C were shifted to 30 °C for 30 min followed by image acquisition for 1–1.5 h with 1 min time intervals at 30 °C. The duration of anaphase of cells with correct and misaligned spindles was determined as the period from the start of rapid spindle elongation (metaphase-to-anaphase transition) to spindle breakdown[46].

**SAC integrity assays.** To assess the integrity of the SAC, log-phase cultures were synchronized in G1 using α-factor. The α-factor was then washed out and cells were released by resuspension in nocodazole containing fresh YPAD media. Samples were collected every 30 min for microscopy and protein extract preparation. Samples for microscopy were fixed in 70% ethanol and nuclei were stained with DAPI. The number of nuclei per cell and budding status of at least 100 cells were recorded. The frequency of large budded cells was plotted.

**Growth assays for genetic interactions.** The impact of gene deletions or ectopic gene expressions (centromeric or 2 μm plasmids) upon the viability of mutants was assessed with a shuffle strain strategy in which the mutant strains are initially complemented by the corresponding wild-type gene on an URA3-based plasmid (pRS316). The genetic interaction was then analysed by testing the growth of the mutant on 5FOA-containing plates, on which only cells that lost the URA3-based plasmid can grow. Genetic interactions with conditional mutants, such as temperature-sensitive mutants and Gal1-KIN4 (or Gal1-BFA1) strains, were evaluated at relevant restrictive temperature and under Gal1 overexpression conditions, respectively. Growth was assayed by performing drop tests in which serial dilutions of cultures were spotted on corresponding agar plates. Briefly, stationary cell cultures grown in appropriate media were diluted in sterile PBS to give a cell density of $2 \times 10^7$ cells per ml. From this initial dilution, serial dilutions (1:10, 1:10$^2$, 1:10$^3$ and 1:10$^4$) were done and were spotted (5–10 μl of each dilution) on appropriate agar plates. Plates were incubated at appropriate temperature for 2–3 days.

***In vitro* kinase assay.** In vitro kinase assays of yeast purified Cdc28-as/Clb2-TAP were performed in a kinase reaction buffer containing 25 mM Hepes, pH 7.4,

150 mM NaCl, 10 mM MgCl$_2$, 0.05 μM ATP and 5 μCi γ-($^{32}$P) ATP per 20 μl kinase reaction volume. Reactions with yeast purified GST-Cdc5 and GST-Cdc5-KD[42] were performed in 50 mM Tris-HCl pH 7.5, 10 mM MgCl2, 1 mM dithiothreitol and 0.05 μM ATP and 5 μCi γ-($^{32}$P) ATP. The substrates were amylose bead bound MBP-Bfa1, MBP-Bfa1-6A or MBP in solution. To block Cdc28-as 0.5 μl of 2.5mM 1NM-PP1 (dissolved in DMSO), or 0.5 μl DMSO alone was added to the kinase assays. Reactions were held at 30 °C for 30 min. Radioactivity was detected using FujiFilm Bas 1,800 II imaging system (FujiFilm, Tokyo, Japan).

**Fluorescence microscopy.** Time-lapse and fluorescence recovery after photo-bleaching experiments were performed using a DeltaVision RT wide-field fluorescence imaging system (Applied Precision, Issaquah, WA, USA) equipped with a quantifiable laser module, an Olympus IX71 microscope with plan-Apo × 100NA 1.4 oil immersion objective (Olympus, Tokyo, Japan), a camera (Photometrics CoolSnap HQ; Roper Scientific, Tucson, AZ, USA) and SoftWoRx software (Applied Precision). Cells were adhered to the bottom of glass-bottom dishes (MatTek, Ashland, MA, USA) using 6% concanavalin A-Type IV (Sigma). Still images of live or fixed cells were acquired using a Zeiss Axiophot microscope equipped with a 100 × NA 1.45 Plan-Fluor oil immersion objective (Zeiss, Jena, Germany), Cascade 1K CCD camera (Photometrics, Tucson, AZ, USA) and MetaMorph software (Universal Imaging Corp., Chesterfield, PA, USA). To score the budding index, cells were fixed with 70% ethanol and resuspended in PBS containing 1 μg ml$^{-1}$ DAPI. The size of the buds and the distribution of DNA-stained regions were counted for 100–150 cells per time point. For visualization of GFP-Tubulin, cells were fixed using 4% paraformaldehyde. Other GFP-tagged proteins were imaged without fixation. Mean fluorescence signal intensities were measured using ImageJ by selecting a fixed size area around the signal of interest at the SPBs. In addition, the background mean signal intensity was measured at an area in the cytoplasm, close to the SPBs. The mean signal intensity at the SPBs was corrected for the background by subtraction of the background signal from the SPB signal. Most data sets that were pairwise compared were normally distributed that can be seen in box-and-whisker plots. All data sets that show significant difference were then confirmed to have a power of >80% (at a significance level of 0.05) using power analysis for two-independent sample t-test. All images were processed in ImageJ (NIH, Bethesda, MD, USA), Adobe Photoshop and Adobe Illustrator (Adobe Systems, San Jose, CA, USA). No manipulations were performed other than brightness, contrast and colour balance adjustments.

**Additional methods.** Additional methods (yeast strains and plasmids, growth conditions, protein methods and genome-wide synthetic lethality screens) are described in the Supplementary Methods.

**Data availability.** Data supporting the findings of this study are available in the article and its Supplementary Information files, or from the corresponding author upon request.

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

## Acknowledgements

We thank Thomas Ruppert (ZMBH-DKFZ Alliance Mass spectrometry facility) for excellent support with MS-analysis and Dorothee Albrecht for technical support. We thank Elmar Schiebel and Marco Geymonat for sharing reagents and/or equipment; and Iain Hagan and Elmar Schiebel for critical reading of the manuscript. This work was supported by the German Research Council (DFG) Grant PE1883-1/2 and Cooperative Grant SFB1036 granted to G.P.; A.K.C. was funded by the DFG Grant PE1883-1/2; R.D.-S. is funded by the SFB1036; B.K. is funded by the SFB873 and a member of the HIBGS international PhD Program of the University of Heidelberg; A.K. and M.K. acknowledge funding of SFB1036; G.P. is supported by the DFG Heisenberg Program. We apologize to those whose work could not be cited because of the limited space.

## Author contributions

A.K.C. and G.P. planned and coordinated the project, and wrote the manuscript. A.K.C. performed most of the experiments. A.K. performed the synthetic genetic array (SGA) screening and participated in manuscript editing. R.D.-S. performed the experiments shown in Figure 6c and Supplementary Figs 6 and 9c,e,f and purification of Bfa1 for MS analysis. B.K. performed the experiments shown in Figure 4d, and Supplementary Figs 8 and 9a. All authors read the manuscript.

## Additional information

**Competing financial interests:** The authors declare no competing financial interests.

**Publisher's note**: 

