## [Peer Review File · Nature Communications]

Reviewer #1 (Remarks to the Author)

This manuscript addresses how mitotic signaling pathways (FEAR, MEN and SPOC) orchestrate chromosome segregation in time and space.

For daughter cells to inherit a complete genomic complement it is essential that the mitotic spindle is properly positioned along the mother-daughter polarity axis. In budding yeast a surveillance mechanism known as the spindle position checkpoint (SPOC) exists that monitors spindle position and inhibits mitotic exit by restraining the activity of the Mitotic Exit Network (MEN) until the mitotic spindle is properly oriented.

Albeit a lot is known about exit from mitosis the molecular mechanism by which the SPOC function remains unclear.

Here the authors shed light into the process. First they highlight a central role for the kinase Kin4 in counterbalancing FEAR-mediated activation of the MEN within the mother cell. Second, they identify Bfa1 and Cdc15 as key targets of the FEAR network. Finally they uncover a FEAR independent control of mitotic exit specific for the daughter compartment that requires Lte1 and Ste20.

Taken together these results allow the authors to propose an elegant model explaining the intricate net of dependency and regulation of the mitotic pathways at the level of compartmentalization and timing.

The manuscript is very well written. The experiments are beautiful and the conclusions supported by the data. The experiments are clear-cut, and the interpretations appropriate. I really enjoyed it.

Reviewer #2 (Remarks to the Author)

In this paper Caydasi and co-workers explore the contribution of the FEAR network in promoting mitotic exit in cells with misaligned spindles and its interconnection with the SPOC. They show that increasing the expression of SPO12 impairs SPOC proficiency and conversely, inhibition of FEAR maintains SPOC activity even in the absence of Kin4. Importantly, in cells with misaligned spindles and active SPOC the FEAR is also active demonstrating that the SPOC does not solely function by inhibiting the FEAR per se (at least not at the level of nuclear release and activity of Cdc14). Indeed, FEAR and SPOC appear to work in parallel: FEAR inactivation does not restore Bfa1 localisation (nor Cdc5-dependent phosphorylation) in $\Delta kin4$ cells with misaligned spindles, despite the ability of those cells to activate the SPOC. The authors propose that the contribution of the FEAR in controlling mitotic exit is exerted on Cdc14-dependent dephosphorylation of two elements, the MEN kinase Cdc15 and the SPOC component Bfa1. The authors use non-phosphorylatable versions of Cdc15 and Bfa1 and they show that, in cells with misaligned spindles, those mutants impair the SPOC irrespective of the FEAR network. The last part of the study shows that Lte1 and Ste20, two known positive regulators of the MEN that are normally localised in the bud compartment, can induce mitotic progression in cells with misaligned spindle (i.e. bypass the SPOC) if they are instead forced to localise at the mother cell cortex. While this was already known for Lte1, the authors show here that this is true even in a $\Delta kin4 \Delta spo12$ strain demonstrating that Lte1 has also a Kin4-independent role in activating the MEN. The authors identify STE20 in a SGA screen for genes essential for viability of a $\Delta spo12 \Delta lte1 \Delta kin4$ strain. The ability of mother-localised Ste20 to induce mitotic progression in $\Delta spo12 \Delta kin4$ with misaligned spindles is new and confirms its role in MEN regulation. The quality of the data presented is of a high standard (very typical of the work done in Pereira's lab) and I think it advances importantly our understanding of the interconnection between FEAR, SPOC and MEN. Therefore I feel the study is suitable for the readership of Nature Communications. However, I would like to recommend a number of revisions as condition for final acceptance.

One of the main points of this paper is the molecular nature of the interaction between the FEAR network and the SPOC in the presence of misaligned spindles. Caydasi and co-workers propose that the de-phosphorylation of Cdc15 (a known FEAR-dependent substrate of Cdc14) and Bfa1 (a new substrate proposed in this manuscript) during early anaphase release of Cdc14 explain bypass of the SPOC arrest in a Δ kin4 strain.

1) My main concern refers to the Bfa1 data:

Bfa1 is a heavily regulated protein, mainly by phosphorylation. In this work a new layer of regulation is proposed. While the in vitro data on the identification of phosphorylated sites by CDK are quite clear, the authors do not provide any evidence that those sites are actually phosphorylated in vivo by the same kinase. In fact, residue S454 is phosphorylated in vivo and in vitro by Cdc5 (Kim et al 2012). In vivo phosphorylation of T340, T465 and T500 has not been previously reported. According to a study not cited in this manuscript, S274 and T288 are phosphorylated in vivo but only T288 phosphorylation is enriched upon arrest by Cdc20 depletion (mitotic, pre-FEAR arrest) (Jones et al. Cell Cycle, 10:3435, 2011). It is therefore essential that the authors offer biochemical data on whether these sites are indeed phosphorylated in vivo and, in turn, constitute targets of Cdc14 during FEAR-dependent release. Importantly, Bfa1 is a known target of Cdc14 following MEN activation. Thus, support for the authors' proposal here requires the demonstration that Bfa1 is subject to such regulation within the timescale of FEAR, for example, by time-course analysis in checkpoint-arrested cells (misoriented spindles) versus cells in which the MEN has been abrogated genetically.

Moreover, the impact of S/T to A substitutions is only assessed genetically leading to the interpretation that Bfa1-6A is impaired in inhibiting Tem1. In the absence of biochemical data, and the fact that S/T to D is not found to be a more potent inhibitor (in fact the opposite as per Figure 6C) it remains questionable if those substitutions are simply impairing Bfa1 function beyond any links to phosphorylation.

Could the authors provide explicit data on mobility shift of Bfa1 versus Bfa1-6A along the cell cycle and in the context of checkpoint activation? Furthermore, do they observe similar changes in cells in which the MEN has been disabled?

Regarding the authors' interpretation on the role of Bfa1 phosphorylation - does the S/T to D mutant behave as a more effective inhibitor of Tem1 thus counteracting more efficiently the phenotype of Spo12 up-regulation described in Figure 1D?

The authors argue that Bfa1 function in the SPOC may be inferred from its ability to support metaphase arrest in response to nocodazole (fig. 6B) for their evaluation of Bfa1-6A. However, SAC-dependent response to nocodazole would hold the FEAR network inactive and it is therefore a less relevant context to extrapolate on Bfa1 status and ability to regulate the MEN upon SPOC challenging by spindle misalignment.

Moreover from figure 6C, third graph, it seems that the biological contribution of bfa1-6A over wild type Bfa1 is otherwise minor when assessed in a Δ spo12 background (compare with data presented in figure 5F for cdc15-7A). This should be clearly discussed in the text.

Finally, the authors should study the localisation of the bfa1-6A, bearing in mind that the S454 position has been implicated in the asymmetric localisation of Bfa1 (Kim et al 2012).

2) I am also concerned regarding the interpretation of the cdc15-7A data:

According to figure 5F, cdc15-7A cells with misaligned spindles bypass SPOC-dependent arrest. However, the authors' comment that "...the dephosphorylation of Cdc15 by the FEAR pathway that promotes mitotic exit of cells with misaligned spindles..." (Page 9, middle paragraph) is somewhat confusing given that the FEAR pathway normally operates in early anaphase irrespective of

whether spindle misalignment may later trigger proficient SPOC arrest. The authors only linked FEAR with mitotic exit in a $\Delta kin4$ background here.

Thus, the (transient?) dephosphorylation of Cdc15 (dependent on FEAR) during a SPOC arrest is not proven in this study but should be demonstrated. The authors cannot exclude other possibilities e.g. Kin4 specifically inhibiting Cdc15 dephosphorylation (hence counteracting FEAR at the level of Cdc15).

3) Regarding the last section of the manuscript, the finding that Lte1 can promote mitotic exit also in a Kin4-independent way is very interesting and re-opens the question of the role of Lte1 as a putative GEF. Unfortunately the manuscript does not shed light on this matter beyond an already proven connection with Ras proteins. In this regard, Ras has been implicated in the localisation of Lte1 at the bud cortex (Yoshida 2003) but in their experiment Caydasi and co-workers infer a more structural role for Ras since Lte1 has been tethered to the mother cell cortex via Sfk1 in the strain under study. Thus, their result should be validated by inactivation of yeast Ras under similar conditions.

If Lte1 has a Kin4-independent role in activating the MEN (at the level of Bfa1/Bub2 or Tem1, as proposed in page 14), how do the authors interpret the fact that mother-localised Lte1 cannot bypass a nocodazole arrest (Bertazzi 2011)?

4) The finding that mother-localised Ste20 promotes mitotic exit in $\Delta spo12 \Delta kin4$ cells with misaligned spindles is also noteworthy. However, this result may be expected given that Hofken & Schiebel 2002 showed that Ste20 is a positive MEN regulator. Unfortunately, like for Lte1, the authors do not present any data to propose a molecular role for the observed bypass of the SPOC.

Therefore, the authors should at least address the following points:

- Is mother-localised Ste20 sufficient to bypass SPOC arrest in SPO12 KIN4 cells with misaligned spindles, as reported for Lte1?
- Furthermore, does mother-localised Ste20 bypass a nocodazole-induced arrest?
- Finally, could the authors consider possible molecular roles for Lte1 and Ste20 in their discussion integrating the findings in the current study?

Overall I think this is an interesting study that merits publication provided that the points I have raised are addressed.

Minor point:

page 9, end of the first paragraph - This sentence is slightly misleading, I think it would be more clear if it included "the lack of FEAR restore the low level of the SPB-bound Cdc15 and Mob1..."

Reviewer #3 (Remarks to the Author)

In their manuscript, Caydasi et al. aim to shed light on the mechanisms by which mitotic exit is regulated in *Saccharomyces cerevisiae*, particularly when equal partitioning of the genomic DNA is compromised by the improper orientation of the mitotic spindle, and consequently the spindle position checkpoint (SPOC) is activated. In budding yeast, mitotic exit is promoted by the nucleolar release of the Cdc14 phosphatase through the combined action of the FEAR and MEN pathways. The authors propose that a central role of the Kin4 kinase, a key SPOC component, is to impede a FEAR-dependent activation of the MEN within the mother cell. This role is supported by the surprising observation that deletion of SPO12, which inactivates the FEAR, makes Kin4 dispensable for SPOC function. Caydasi et al. further suggest that, in the absence of Kin4, the FEAR promotes inappropriate mitotic exit in cells with mispositioned spindles by facilitating the Cdc14-dependent dephosphorylation of the MEN inhibitor Bfa1 and the MEN kinase Cdc15. Finally,

the authors propose that mitotic exit is not triggered in cells lacking both Kin4 and Spo12 until the spindle is properly aligned and one of the spindle pole bodies (SPBs) enters the daughter cell because MEN signaling specifically requires the bud-confined proteins Lte1 and Ste20, which together constitute a second layer of regulation that allows the spatial compartmentalization of mitotic exit.

The restitution of the SPOC function in cells lacking Kin4 by means of the inactivation of the FEAR pathway is a highly interesting result, and it is well supported by the data provided by the authors. The experiments described in the manuscript were neatly executed and the results well presented. The included diagrams facilitate the understanding of the different pathways and the authors' conclusions. However, I believe that the proposed mechanism by which lack of FEAR facilitates restitution of the functionality of the SPOC in kin4Δ cells requires further support. In this sense, and even though the authors make an effort to discard that the FEAR could promote the Cdc5-dependent inhibition of Bfa1/Bub2, I am concerned as to whether the experiments presented in the manuscript can undoubtedly eliminate this possibility as the most feasible mechanism. Additionally, it is not clear to me whether the FEAR-dependent Cdc15 activation is a main driver of the SPOC defects in kin4Δ cells. A more conclusive demonstration of the exact mechanism by which the FEAR rescues the SPOC deficiency in the absence of Kin4 would strengthen the manuscript, and will definitely contribute to facilitate our understanding of how mitotic exit is prevented until the spindle is properly oriented and the correct distribution of the chromosomes between the mother and the daughter cell is secured.

My major points regarding the manuscript are the following:

1.- The fact that deletion of SPO12 does not recover the SPOC deficiency in cells lacking Bfa1 or Bub2 (Figures 2C and S2C-D) is hard to reconcile with the fact that the authors later propose that FEAR induces premature mitotic exit in cells with mispositioned spindles that lack the SPOC by activating Cdc15, which acts downstream of Tem1 in MEN signaling. I would expect to see at least a partial rescue of the SPOC function, as observed in the cdc15-7A mutant. This result suggests that the most important aspect of the FEAR that affects the functionality of the SPOC is its ability to somehow facilitate an inhibition of Bfa1/Bub2 activity, which is mainly accomplished by the Cdc5-dependent phosphorylation of the GAP in anaphase. An inefficient Cdc5-dependent inhibition of Bfa1 in cells that lack FEAR could easily explain why SPO12 deletion can rescue the SPOC deficiency in kin4Δ cells, which can no longer block inactivation of Bfa1 by Cdc5, but not in a bfa1Δ mutant.

2.- In order to exclude that FEAR promotes the Cdc5-dependent inhibition of Bfa1, the authors check the phosphorylation status of Bfa1 (Figure 4A) and other substrates of the Polo-like kinase at the SPBs (Figure S5). However, there are many phosphorylation events taking place on Bfa1, some Cdc5-dependent and some independent, which complicates the interpretation of the results in a SDS-PAGE. The analysis is further complicated by the fact that some of the strains are SPOC-deficient and therefore do not accumulate in anaphase. In the experiment shown in Figure 4, the slowest migrating form of Bfa1, which corresponds to the Cdc5-dependent phosphorylation (see KIN4 SPO12 strain in Figure 4A), seems to show up later and it is less intense in kin4Δ spo12Δ cells than in the kin4Δ SPO12 mutant. Ideally, the experiment should have been carried out in a cdc15 conditional mutant background, to prevent mitotic exit and accumulate Bfa1 in its most phosphorylated form. This could help to normalize possible differences among strains. On a different note, it is worth noting that the fact that Cdc5 phosphorylates other substrates at the SPBs in a kin4Δ spo12Δ mutant does not necessarily exclude that the Cdc5-dependent phosphorylation of Bfa1 could be specifically affected.

3.- The authors propose that SPO12 deletion promotes SPOC arrest through inhibition of Cdc15. Although it has been demonstrated that Cdc14 released by the FEAR enhances MEN signaling by dephosphorylating Cdc15 and Mob1, which promotes their binding to the SPBs, it has also been shown that Tem1 activity is required to recruit Cdc15 to the SPBs. The fact that deletion of SPO12

prevented accumulation of Cdc15 on the SPBs in *kin4Δ* cells with mispositioned spindles (Figure 5E) would also be expected if Tem1 were inactive due to the inhibitory action of Bfa1. Thus, and again, this result does not allow excluding that what the FEAR mainly does is to facilitate Bfa1 inhibition, and consequently Tem1 activation. Along the same lines, the fact that a *cdc15-7A* allele promotes mitotic exit of cells with misaligned spindles both in the presence and the absence of the FEAR is completely expected, since the MEN is being activated downstream of Tem1. In fact, that fewer cells with a SPOC-deficient phenotype accumulated in *kar9Δ cdc15-7A spo12Δ* cells than in *kar9 cdc15-7A* (Figure 5F) is also in agreement with the possibility that FEAR is enabling the inhibitory action of Cdc5 on Bfa1, since lack of Spo12 would promote Tem1 inactivation, and therefore Cdc15 loading on the SPBs and MEN signaling would be less facilitated.

4.-The authors suggest that Bfa1 phosphorylation by CDK is required to engage the SPOC arrest, and that lack of Spo12 in *kin4Δ* cells impedes the Cdc14-dependent dephosphorylation of Cdk-dependent sites on Bfa1, thereby rescuing SPOC function. To test their hypothesis, Caydasi et al. generated a mutant version of Bfa1 in which six Cdk consensus sites in the protein are mutated to alanine (Bfa1-6A). However, there are several issues regarding the Bfa1-6A protein. First of all, the authors show that Bfa1-6A cannot be phosphorylated *in vitro* by Clb2-Cdc28 (Figure 6A). However, Bfa1-6A seems to be phosphorylated *in vivo* to a similar level than wild type Bfa1 (Figure S6B). A more careful analysis of the phosphorylation status of the mutant Bfa1 proteins (similar to that shown in Figure 4A) would have been more appropriate to assess the *in vivo* consequences of the mutations. Furthermore, it should be also checked whether these mutations affect SPB localization of Bfa1/Bub2, which is another important aspect of the regulation of the GAP. Finally, it is worth noting that phosphorylation of a protein by Cdc5 usually requires the priming phosphorylation by CDK of a Polo-binding site within the protein. Interestingly, the sequence of Bfa1 has a consensus Polo-binding site that includes amino acids S453 and S454, one of the amino acids mutated in Bfa1-6A. In fact, S454 is one of the four residues mutated in the Bfa14A mutant described by Kim et al., which cannot be phosphorylated by Cdc5 (Kim et al., 2012). Is the Bfa1-6A mutant still phosphorylated by Cdc5 in an *in vitro* assay similar to that shown in Figure 6A? If phosphorylation of the Bfa1-6A mutant by Cdc5 were affected, this would complicate the evaluation of the effects of the mutant on SPOC functionality.

Finally, some minor points are:

5.- Overexpression of SPO12 determines improper mitotic exit in cells with mispositioned spindles, while lack of Spo12 rescues the functionality of the SPOC in *kin4Δ* cells. However, this does not mean that the deficiency or the overexpression of Spo12 affect mitotic exit signaling in cells with mispositioned spindles merely due to opposite mechanisms. As such, and while overexpression of SPO12 suppresses the lethality caused by Bfa1 overexpression, lack of Spo12 does not rescue the SPOC deficiency in cells lacking BFA1. Therefore, and although the link between FEAR and the SPOC is later clearly established, the initial overexpression experiments do not really provide a basis for a possible role of Spo12 in the SPOC. On a different note, the fact that overexpression of Spo12 rescues the temperature sensitivity of many MEN mutants, but not of cells carrying *cdc14-ts* alleles, had been previously well established in the literature (e.g., see Jaspersen et al., 1998), so Figure S1B is not really required.

6.- In Page 9, it is stated that "[...] Thus, lack of FEAR restores the SPB-bound Cdc15 and Mob1 levels in *kin4Δ* cells with misaligned spindles to prevent MEN activation". However, this sentence leads to confusion, since deletion of SPO12 in fact prevents Mob1-GFP and Cdc15-GFP loading on the SPBs in *kin4Δ* cells (Figures 5C-E). The sentence should be rephrase to avoid this confusion.

Answer to reviewers

We would like to thank all three reviewers for the constructive comments on our manuscripts.

Reviewer #2 (Remarks to the Author):

In this paper Caydasi and co-workers explore the contribution of the FEAR network in promoting mitotic exit in cells with misaligned spindles and its interconnection with the SPOC. They show that increasing the expression of SPO12 impairs SPOC proficiency and conversely, inhibition of FEAR maintains SPOC activity even in the absence of Kin4. Importantly, in cells with misaligned spindles and active SPOC the FEAR is also active demonstrating that the SPOC does not solely function by inhibiting the FEAR per se (at least not at the level of nuclear release and activity of Cdc14). Indeed, FEAR and SPOC appear to work in parallel: FEAR inactivation does not restore Bfa1 localisation (nor Cdc5-dependent phosphorylation) in $\Delta kin4$ cells with misaligned spindles, despite the ability of those cells to activate the SPOC. The authors propose that the contribution of the FEAR in controlling mitotic exit is exerted on Cdc14-dependent dephosphorylation of two elements, the MEN kinase Cdc15 and the SPOC component Bfa1. The authors use non-phosphorylatable versions of Cdc15 and Bfa1 and they show that, in cells with misaligned spindles, those mutants impair the SPOC irrespective of the FEAR network. The last part of the study shows that Lte1 and Ste20, two known positive regulators of the MEN that are normally localised in the bud compartment, can induce mitotic progression in cells with misaligned spindle (i.e. bypass the SPOC) if they are instead forced to localise at the mother cell cortex. While this was already known for Lte1, the authors show here that this is true even in a $\Delta kin4 \Delta spo12$ strain demonstrating that Lte1 has also a Kin4-independent role in activating the MEN. The authors identify STE20 in a SGA screen for genes essential for viability of a $\Delta spo12 \Delta lte1 \Delta kin4$ strain. The ability of mother-localised Ste20 to induce mitotic progression in $\Delta spo12 \Delta kin4$ with misaligned spindles is new and confirms its role in MEN regulation. The quality of the data presented is of a high standard (very typical of the work done in Pereira's lab) and I think it advances importantly our understanding of the interconnection between FEAR, SPOC and MEN. Therefore I feel the study is suitable for the readership of Nature Communications. However, I would like to recommend a number of revisions as condition for final acceptance.

One of the main points of this paper is the molecular nature of the interaction between the FEAR network and the SPOC in the presence of misaligned spindles. Caydasi and co-workers propose that the de-phosphorylation of Cdc15 (a known FEAR-dependent substrate of Cdc14) and Bfa1 (a new substrate proposed in this manuscript) during early anaphase release of Cdc14 explain bypass of the SPOC arrest in a $\Delta kin4$ strain.

1) My main concern refers to the Bfa1 data:

Bfa1 is a heavily regulated protein, mainly by phosphorylation. In this work a new layer of regulation is proposed. While the *in vitro* data on the identification of phosphorylated sites by CDK are quite clear, the authors do not provide any evidence that those sites are actually phosphorylated *in vivo* by the same kinase. In fact, residue S454 is phosphorylated *in vivo* and *in vitro* by Cdc5 (Kim et al 2012). *In vivo* phosphorylation of T340, T465 and T500 has not been previously reported. According to a study not cited in this manuscript, S274 and T288 are phosphorylated *in vivo* but only T288 phosphorylation is enriched upon arrest by Cdc20 depletion (mitotic, pre-FEAR arrest) (Jones et al. Cell Cycle, 10:3435, 2011). It is therefore essential that the authors offer biochemical data on whether these sites are indeed phosphorylated *in vivo* and, in turn, constitute targets of Cdc14 during FEAR-dependent release.

We now performed MS-analysis of Bfa1 enriched from metaphase-arrested cells (Cdc20-depletion). Our analysis identified *in vivo* phosphorylation of residues T288, S454, T340 and T500 (Supplementary Fig. 9). T340 and T500 were not reported before. We could not identify

phosphorylation of residues S274 and T465. Of those, S274 was previously reported to be phosphorylated in cells arrested in the G1-phase but not mitosis (Jones et al., Cell Cycle 2011). Phosphorylation of T465 was not detected in previous studies. We therefore considered that T288, S454, T340 and T500 are the major mitotic Cdk sites that are phosphorylated in vivo in Bfa1. Accordingly, we constructed the Bfa1-4A mutant, in which these sites were mutated to alanine. We compared the SPOC proficiency of Bfa1-4A and Bfa1-6A. We now show that Bfa1-4A is SPOC deficient (Figure 6i). This implies that T288, S454, T340 and T500 are the major Cdk sites that contribute to Bfa1 function in SPOC.

Importantly, Bfa1 is a known target of Cdc14 following MEN activation. Thus, support for the authors' proposal here requires the demonstration that Bfa1 is subject to such regulation within the timescale of FEAR, for example, by time-course analysis in checkpoint-arrested cells (misoriented spindles) versus cells in which the MEN has been abrogated genetically.

We analysed Bfa1 phosphorylation in a MEN block (Tem1 depletion) in the presence or absence of FEAR (*spo12Δ*). Upon Tem1 depletion, both *SPO12* and *spo12Δ* cells arrest in late mitosis with hyperphosphorylated Bfa1 (Fig. 6a, 6c and Supplementary Fig. 9e). However, Bfa1 migrates even slower in *spo12Δ* cells than in *SPO12* cells (Fig 6a). In addition we observed that hypophosphorylated forms of Bfa1 appeared in *SPO12* but not in *spo12Δ* cells (Figure 6a and Supplementary Fig. 9e, t 90, 105 min). Thus we were able to conclude that Bfa1 becomes dephosphorylated by FEAR-released Cdc14. Whether FEAR contributes to Bfa1 dephosphorylation during spindle misalignment (Figure 4) was not straightforward to address. In both *SPO12* and *spo12Δ* cells, Bfa1 remains hypophosphorylated, making it difficult to conclude about the FEAR contribution to Bfa1 phosphorylation under this condition. We were able to see a difference in the migration profile of hyperphosphorylated Bfa1 in the absence and presence of FEAR, but we do not know if any difference in phosphorylation profile of hypophosphorylated Bfa1 in the absence and presence of FEAR could result in a shift in the migration profile of Bfa1 on SDS-PAGE. We furthermore tried to assess the Cdk phosphorylation of Bfa1 by pulling down Bfa1 and detecting the phosphorylation using commercially available antibodies against proline directed kinases, however those antibodies failed to detect specific phospho-bands in our experimental setup. The fact that Bfa1-6A causes SPOC deficiency (also in *spo12Δ kin4Δ* cells) indicates that dephosphorylation of Bfa1 at Cdk sites facilitates mitotic exit in late anaphase. Our genetic and Bfa1-6A localisation data indicate that Cdk and Kin4 may work in parallel to keep Bfa1 active (Figs. 6 and S9a). Yet we cannot exclude the possibility that Bfa1 dephosphorylation by FEAR-released Cdc14 might be prevented during spindle misalignment to engage the SPOC by an unknown mechanism independent of Kin4. This possibility is now discussed (page 16).

Moreover, the impact of S/T to A substitutions is only assessed genetically leading to the interpretation that Bfa1-6A is impaired in inhibiting Tem1. In the absence of biochemical data, and the fact that S/T to D is not found to be a more potent inhibitor (in fact the opposite as per Figure 6C) it remains questionable if those substitutions are simply impairing Bfa1 function beyond any links to phosphorylation.

Bfa1 is required for both SAC and SPOC arrest. The genetic evidences strongly support that Bfa1-6A is not able to support SPOC, yet Bfa1-6A supports the metaphase arrest induced by nocodazole (SAC-dependent). This shows that, the serine to alanine substitutions in Bfa1 do not impair Bfa1 function in general, which would be expected if these substitutions would influence Bfa1 protein stability or folding. Indeed, Bfa1-6A is expressed at the same level as Bfa1 (Supplementary Fig 9d). The localisation of Bfa1-6A is also similar to Bfa1 (Supplementary Fig 9a). These data thus indicate that phosphorylation of Bfa1 at Cdk sites is specifically required for Bfa1's function in SPOC but not in SAC. Because Bfa1-Bub2 keeps the SPOC active via Tem1 inhibition, we think is valid to conclude that Bfa1-6A is unable to keep Tem1 inactive during the SPOC arrest.

To investigate whether Bfa1-6D is a more potent MEN inhibitor, we analysed whether Bfa1-6D would impair the growth of MEN loss-of-function mutants. The growth of MEN temperature sensitive mutants (Supplementary Fig. 9g) was not affected by Bfa1-6D. However, we

observed that Bfa1-6D impaired cell growth of Cdc5-temperature sensitive mutants, suggesting that Bfa1-6D is a more potent inhibitor of MEN when Bfa1-Bub2 GAP complex inactivation is compromised. In agreement with this conclusion, Bfa1-6D also impaired the growth of *rts1Δ* cells overexpressing *KIN4* (a condition in which Kin4 is not fully active and therefore does not completely block MEN activation (Chan and Amon, Genes&Dev. 2009, Caydasi et al., JCB 2010). These data are now shown in Fig. 6f-6g.

Could the authors provide explicit data on mobility shift of Bfa1 versus Bfa1-6A along the cell cycle and in the context of checkpoint activation? Furthermore, do they observe similar changes in cells in which the MEN has been disabled?

We now included the data showing the mobility shift of Bfa1 and Bfa1-6A during the cell cycle progression (Supplementary Fig 9c), during cell cycle arrest upon MEN inactivation by Tem1-depletion (Fig. 6c and Supplementary Fig 9e) and upon checkpoint activation (Supplementary Fig. 9f). Similar to our observations of Bfa1 phosphorylation in *spo12Δ* cells, in cells with properly aligned spindles and inactive MEN (G1-arrest and release under conditions that allow Tem1-depletion), *Bfa1-6D* migrated slower than *BFA1* or *Bfa1-6A* (Supplementary Fig. 9e). The same was true in cells with misaligned spindles (Supplementary Fig. 9f). However, unlike Cdc5, we could not assign a specific band to Cdk-phosphorylated Bfa1.

Regarding the authors' interpretation on the role of Bfa1 phosphorylation - does the S/T to D mutant behave as a more effective inhibitor of Tem1 thus counteracting more efficiently the phenotype of Spo12 up-regulation described in Figure 1D?

We observed no significant difference in the growth of *BFA1* and *bfa1-6D* cells overexpressing *SPO12* (Figure A, see below), although there was a slight reduction in average % of SPOC deficient phenotype. However, *bfa1-6D* impaired the growth of *cdc5-10* and *rts1Δ Gal1-KIN4* cells (Fig. 6f-g), implying that it can work as a more potent inhibitor of mitotic exit under these conditions.

Figure A. Effect of 2μm-SPO12 on SPOC deficiency of the indicated cells.

The authors argue that Bfa1 function in the SPOC may be inferred from its ability to support metaphase arrest in response to nocodazole (fig. 6B) for their evaluation of Bfa1-6A. However, SAC-dependent response to nocodazole would hold the FEAR network inactive and it is therefore a less relevant context to extrapolate on Bfa1 status and ability to regulate the MEN upon SPOC challenging by spindle misalignment.

Sorry for the confusion. We did not perform this experiment to study Bfa1 function in SPOC. For SPOC analysis, we investigated *bfa1-6A kar9Δ* cells with misaligned spindles (Fig 6). We have used the nocodazole treatment to test the functionality of Bfa1-6A. It is well-established that the Bfa1-Bub2 GAP complex is required for the SAC-induced arrest. In cells lacking Bfa1, cells treated with nocodazole exit mitosis due to MEN activation. This shows that a functional Bfa1 is required for SAC. Because Bfa1-6A can hold the SAC arrest (unlike *bfa1Δ* cells), we conclude that the alanine substitutions do not abrogate Bfa1 function in general (e.g. through structural defects).

Moreover from figure 6C, third graph, it seems that the biological contribution of *bfa1-6A* over

wild type Bfa1 is otherwise minor when assessed in a $\Delta spo12$ background (compare with data presented in figure 5F for *cdc15-7A*). This should be clearly discussed in the text.

Thank you for pointing this out. We now discuss that unlike *Cdc15-7A*, Bfa1-6A is not able to promote mitotic exit of cells with misaligned spindles in the absence of FEAR (pages 11 and 16). This difference could be explained by the fact that Bfa1-6A is still under Kin4 control, as deletion of *KIN4* in Bfa1-6A *spo12 Δ kin4 Δ* cells caused SPOC deficiency, as well as the fact that *cdc15-7A* is a more potent MEN activator (pages 11 and 16-17).

Finally, the authors should study the localisation of the bfa1-6A, bearing in mind that the S454 position has been implicated in the asymmetric localisation of Bfa1 (Kim et al 2012).

Thank you for raising this point. Kim et al (Plos Genetics, 2012) showed that the percentage of late anaphase cells (*cdc15-ts* arrested cells) with Bfa1-S454A-GFP at the mother SPB was approximately 10% higher in comparison to *BFA1-GFP* expressing cells. The decrease in Bfa1 asymmetry (e.g. detectable Bfa1 at the mother SPB in late anaphase) increased in Bfa1-3A (S452A, S453A and S454A) and Bfa1-4A (S452A, S453A, S454A, S559A), suggesting that S454A alone does not drastically influence Bfa1 SPB binding. We now analysed the SPB localisation of Bfa1 and Bfa1-6A in late anaphase arrested cells with normal or misaligned spindles. We could not detect differences in the localisation profile of Bfa1 and Bfa1-6A in our strain background (Supplementary Fig. 9a-b). Bfa1 and Bfa1-6A associated preferentially in almost 100% of the cells at only the daughter cell SPB (Supplementary Fig. 9b).

2) I am also concerned regarding the interpretation of the *cdc15-7A* data: According to figure 5F, *cdc15-7A* cells with misaligned spindles bypass SPOC-dependent arrest. However, the authors' comment that "...the dephosphorylation of Cdc15 by the FEAR pathway that promotes mitotic exit of cells with misaligned spindles..." (Page 9, middle paragraph) is somewhat confusing given that the FEAR pathway normally operates in early anaphase irrespective of whether spindle misalignment may later trigger proficient SPOC arrest. The authors only linked FEAR with mitotic exit in a $\Delta kin4$ background here.

Thank you for pointing this out. We now changed this passage to avoid confusion (now page 10 first sentence).

Thus, the (transient?) dephosphorylation of Cdc15 (dependent on FEAR) during a SPOC arrest is not proven in this study but should be demonstrated. The authors cannot exclude other possibilities e.g. Kin4 specifically inhibiting Cdc15 dephosphorylation (hence counteracting FEAR at the level of Cdc15).

We now show that Cdc15 hypophosphorylated forms accumulate in cells with misaligned spindles (Supplementary Fig. 6), hence Kin4 likely does not specifically inhibit Cdc15 dephosphorylation by FEAR. This is rather consistent with a model in which cells require Kin4 to regulate Bfa1-Bub2 to efficiently keep Tem1 inactive, thereby preventing SPB localisation of Cdc15. This assumption is in agreement with the fact that Cdc15 and Mob1 accumulate at the SPB in *kin4 Δ* cells at higher levels than in *spo12 Δ kin4 Δ* cells (Fig. 5). However, we cannot exclude that Kin4 might also prevent MEN activation downstream of Cdc15 in cells with misaligned spindles.

3) Regarding the last section of the manuscript, the finding that Lte1 can promote mitotic exit also in a Kin4-independent way is very interesting and re-opens the question of the role of Lte1 as a putative GEF. Unfortunately the manuscript does not shed light on this matter beyond an already proven connection with Ras proteins. In this regard, Ras has been implicated in the localisation of Lte1 at the bud cortex (Yoshida 2003) but in their experiment Caydasi and co-workers infer a more structural role for Ras since Lte1 has been tethered to the mother cell cortex via Sfk1 in the strain under study. Thus, their result should be validated by inactivation of yeast Ras under similar conditions.

We now analysed the ability of Lte1 to promote mitotic exit when targeted to the mother cell

compartment in *ras1Δ/ras2Δ* cells. We show that mother-cell-enriched Lte1 triggers mitotic exit in the presence or absence of Ras1/Ras2 (Fig. 7d). Mitotic exit seemed less efficient in the absence of Ras1/2, as fewer cells with SPOC deficient phenotypes were observed. This data suggest that if Ras has a structural function it should be very minor because Lte1 is still able to cause SPOC deficiency in the absence of Ras1/2. The role of Ras1/2 could be more complex. Therefore, we decided to show this data without making a strong conclusion.

If Lte1 has a Kin4-independent role in activating the MEN (at the level of Bfa1/Bub2 or Tem1, as proposed in page 14), how do the authors interpret the fact that mother-localised Lte1 cannot bypass a nocodazole arrest (Bertazzi 2011)?

This is a good question. This experiment suggests that Lte1, and also Ste20 (see new Supplemental Fig 10f) is only able to support MEN activation after the metaphase to anaphase transition. One possibility is that Lte1 and Ste20 can only activate MEN after FEAR activation that takes place at anaphase onset. However, we consider this possibility unlikely as Lte1 or Ste20 is able to promote mitotic exit in cells without FEAR (e.g. mother cell recruitment of Lte1 or Ste20 causes mitotic exit of *spo12Δ* cells with misaligned spindles). Therefore, it is possible that Lte1 and Ste20 require another event at the metaphase to anaphase transition. As APC-Cdc20 becomes active during this cell cycle transition, it is tempting to speculate that APC-Cdc20 may trigger protein degradation of a yet to be identified mitotic inhibitor, which would counteract the function of Lte1 and Ste20 in mitotic exit. We now discuss this possibility in discussion.

4) The finding that mother-localised Ste20 promotes mitotic exit in $\Delta spo12\Delta kin4$ cells with misaligned spindles is also noteworthy. However, this result may be expected given that Hofken & Schiebel 2002 showed that Ste20 is a positive MEN regulator. Unfortunately, like for Lte1, the authors do not present any data to propose a molecular role for the observed bypass of the SPOC.

Therefore, the authors should at least address the following points:

- Is mother-localised Ste20 sufficient to bypass SPOC arrest in *SPO12 KIN4* cells with misaligned spindles, as reported for Lte1?

Mother cell-localised Ste20 bypasses the SPOC in *SPO12 KIN4 kar9Δ* cells (data shown in figure 8g).

- Furthermore, does mother-localised Ste20 bypass a nocodazole-induced arrest?

Like Lte1, mother-localised Ste20 could not bypass the SAC (data shown in Supplementary Fig. 10f).

- Finally, could the authors consider possible molecular roles for Lte1 and Ste20 in their discussion integrating the findings in the current study?

We now expanded the discussion on the role of Lte1 and Ste20 in mitotic exit activation to include our current data.

Overall I think this is an interesting study that merits publication provided that the points I have raised are addressed.

Minor point:

page 9, end of the first paragraph - This sentence is slightly misleading, I think it would be more clear if it included "the lack of FEAR restore the low level of the SPB-bound Cdc15 and Mob1..."

We changed the sentence as suggested.

Reviewer #3 (Remarks to the Author):

In their manuscript, Caydasi et al. aim to shed light on the mechanisms by which mitotic exit is regulated in *Saccharomyces cerevisiae*, particularly when equal partitioning of the genomic DNA is compromised by the improper orientation of the mitotic spindle, and consequently the spindle position checkpoint (SPOC) is activated. In budding yeast, mitotic exit is promoted by the nucleolar release of the Cdc14 phosphatase through the combined action of the FEAR and MEN pathways. The authors propose that a central role of the Kin4 kinase, a key SPOC component, is to impede a FEAR-dependent activation of the MEN within the mother cell. This role is supported by the surprising observation that deletion of SPO12, which inactivates the FEAR, makes Kin4 dispensable for SPOC function. Caydasi et al. further suggest that, in the absence of Kin4, the FEAR promotes inappropriate mitotic exit in cells with mispositioned spindles by facilitating the Cdc14-dependent dephosphorylation of the MEN inhibitor Bfa1 and the MEN kinase Cdc15. Finally, the authors propose that mitotic exit is not triggered in cells lacking both Kin4 and Spo12 until the spindle is properly aligned and one of the spindle pole bodies (SPBs) enters the daughter cell because MEN signaling specifically requires the bud-confined proteins Lte1 and Ste20, which together constitute a second layer of regulation that allows the spatial compartmentalization of mitotic exit.

The restitution of the SPOC function in cells lacking Kin4 by means of the inactivation of the FEAR pathway is a highly interesting result, and it is well supported by the data provided by the authors. The experiments described in the manuscript were neatly executed and the results well presented. The included diagrams facilitate the understanding of the different pathways and the authors' conclusions. However, I believe that the proposed mechanism by which lack of FEAR facilitates restitution of the functionality of the SPOC in *kin4Δ* cells requires further support. In this sense, and even though the authors make an effort to discard that the FEAR could promote the Cdc5-dependent inhibition of Bfa1/Bub2, I am concerned as to whether the experiments presented in the manuscript can undoubtedly eliminate this possibility as the most feasible mechanism. Additionally, it is not clear to me whether the FEAR-dependent Cdc15 activation is a main driver of the SPOC defects in *kin4Δ* cells. A more conclusive demonstration of the exact mechanism by which the FEAR rescues the SPOC deficiency in the absence of Kin4 would strengthen the manuscript, and will definitely contribute to facilitate our understanding of how mitotic exit is prevented until the spindle is properly oriented and the correct distribution of the chromosomes between the mother and the daughter cell is secured.

My major points regarding the manuscript are the following:

1.- The fact that deletion of SPO12 does not recover the SPOC deficiency in cells lacking Bfa1 or Bub2 (Figures 2C and S2C-D) is hard to reconcile with the fact that the authors later propose that FEAR induces premature mitotic exit in cells with mispositioned spindles that lack the SPOC by activating Cdc15, which acts downstream of Tem1 in MEN signaling. I would expect to see at least a partial rescue of the SPOC function, as observed in the *cdc15-7A* mutant. This result suggests that the most important aspect of the FEAR that affects the functionality of the SPOC is its ability to somehow facilitate an inhibition of Bfa1/Bub2 activity, which is mainly accomplished by the Cdc5-dependent phosphorylation of the GAP in anaphase. An inefficient Cdc5-dependent inhibition of Bfa1 in cells that lack FEAR could easily explain why SPO12 deletion can rescue the SPOC deficiency in *kin4Δ* cells, which can no longer block inactivation of Bfa1 by Cdc5, but not in a *bfa1Δ* mutant.

This is an important point. We agree that an inefficient Bfa1 phosphorylation by Cdc5 would be a good explanation for the SPOC proficiency of *kin4Δ spo12Δ* cells. We tested this possibility by several means. First, we excluded the possibility that in cells lacking FEAR, Cdc5 would be inactive at the SPB outer plaque. We show that Cdc5 phosphorylates the SPB outer plaque components Spc72 and Nud1 in the absence of FEAR (Supplementary Fig. 5b). Second, we show that Bfa1 becomes hyperphosphorylated in both *kin4Δ spo12Δ* and *kin4Δ* cells (Fig. 4) during spindle misalignment. The depletion of Cdc5 in *kin4Δ spo12Δ* cells reduced Bfa1 hyperphosphorylation (this data is now included in Fig. 4d), confirming Cdc5-dependency. Together, this data strongly suggest that the reason by which FEAR deletion restores SPOC in *kin4Δ* cells is unrelated to the Cdc5-dependent regulation of Bfa1.

We now tested this possibility by other means. We hypothesised that, if Cdc5 cannot properly inhibit Bfa1 in the absence of FEAR, we should be able to induce SPOC deficiency by forcing Cdc5 to associate with SPBs both in wild type and in *spo12Δ* cells. We employed the GFP-binder (GBP-strategy) to test this possibility. We targeted Cdc5-GFP to the SPB outer plaque by expressing it in cells carrying Spc42-GBP. As expected, the targeting of Cdc5-GFP to SPBs caused SPOC deficiency and Bfa1 hyperphosphorylation in *kar9Δ* cells (Fig. 4e and 4f). However, Cdc5-GFP was not able to cause SPOC deficiency in *kar9Δ spo12Δ* cells, despite of the fact that Bfa1 also became hyperphosphorylated in those cells. Therefore, our data argues against the hypothesis that the absence of FEAR reinforces SPOC by preventing Cdc5 to phosphorylate Bfa1 at the SPB outer plaque.

2.- In order to exclude that FEAR promotes the Cdc5-dependent inhibition of Bfa1, the authors check the phosphorylation status of Bfa1 (Figure 4A) and other substrates of the Polo-like kinase at the SPBs (Figure S5). However, there are many phosphorylation events taking place on Bfa1, some Cdc5-dependent and some independent, which complicates the interpretation of the results in a SDS-PAGE. The analysis is further complicated by the fact that some of the strains are SPOC-deficient and therefore do not accumulate in anaphase. In the experiment shown in Figure 4, the slowest migrating form of Bfa1, which corresponds to the Cdc5-dependent phosphorylation (see KIN4 SPO12 strain in Figure 4A), seems to show up later and it is less intense in *kin4Δ spo12Δ* cells than in the *kin4Δ SPO12* mutant. Ideally, the experiment should have been carried out in a *cdc15* conditional mutant background, to prevent mitotic exit and accumulate Bfa1 in its most phosphorylated form. This could help to normalize possible differences among strains. On a different note, it is worth noting that the fact that Cdc5 phosphorylates other substrates at the SPBs in a *kin4Δ spo12Δ* mutant does not necessarily exclude that the Cdc5-dependent phosphorylation of Bfa1 could be specifically affected.

In Figure 4, we show Bfa1 phosphorylation profile in cell cycle-synchronised cultures (alpha-factor arrested and released in the presence of auxin to degrade dynein). The budding index profile (Fig. 4b) shows the cell cycle progression and indicates the time that *KIN4*-deleted cells escapes the mitotic arrest. The use of cell cycle synchronised cultures have allowed us to compare the phosphorylation profile of Bfa1 even in strains that are SPOC deficient. Small differences in cell cycle progression may influence the amount of hyperphosphorylated forms of Bfa1 detected in the immunoblots, explaining why we see less hyperphosphorylated Bfa1 in *kin4Δ spo12Δ* cells in comparison to *kin4Δ SPO12* cells at 60 min. Our aim in Figure 4a was not to compare the timing of Bfa1 phosphorylation but the profile of Bfa1 phosphorylation during the time course. We observed a similar profile of Bfa1 hyperphosphorylation at the time that the percentage of mitotic cells reached almost their maximum (80 min). This profile was quite different to wild type or *spo12Δ* cells, indicating that Bfa1 becomes hyperphosphorylated in *kin4Δ spo12Δ* and *kin4Δ* cells. We now show that the hyperphosphorylation profile of Bfa1 is also similar in Cdc5-GFP Spc42-GBP cells in the presence or absence of Spo12 (Fig. 4e).

As suggested by the reviewer, we now show Bfa1 phosphorylation in MEN defective cells (Tem1-depletion, Supplementary Fig. 8e). In both *SPO12* and *spo12Δ* cells, Cdc5-dependent, hyperphosphorylated forms of Bfa1 accumulate in late mitosis. We consider it as unlikely that FEAR inhibits the phosphorylation of Bfa1 by Cdc5 in cells with misaligned or properly aligned spindles as we also stated above as an answer to the first major point of this reviewer.

We agree that the fact that Cdc5 phosphorylates Spc72 and Nud1 at the SPB outer plaque does not exclude that phosphorylation of Bfa1 could be specifically affected by FEAR. In fact, others and we have shown that Kin4 specifically inhibits Cdc5 phosphorylation of Bfa1 but not of Spc72 and Nud1 (Maekawa et al., JCB 2007). However, we detected no difference between Bfa1 phosphorylation profile and SPB binding behaviour in *kin4Δ* and *kin4Δ spo12Δ* cells. Furthermore, as discussed above, deletion of FEAR was still able to revert SPOC deficiency but not Bfa1 hyperphosphorylation triggered by SPB-targeted Cdc5 (Fig. 4e-4f). Yet, we find it necessary to show that FEAR does not influence the regulation of SPB outer plaque components by Cdc5, as this might have indirectly affected mitotic exit.

Therefore, our data indicates that Cdc5 phosphorylates Bfa1 in absence of FEAR and Kin4. We concluded that FEAR rescues SPOC deficiency of *kin4Δ* cells by means that are unrelated to Cdc5-dependent phosphorylation of Bfa1. Because Cdk phosphorylates Bfa1 and Bfa1 non-phosphorylatable mutants are SPOC deficient, we favor the model that Kin4 and Cdk are required to keep the Bfa1-Bub2 GAP complex active in late anaphase to provide SPOC arrest.

3.- The authors propose that SPO12 deletion promotes SPOC arrest through inhibition of Cdc15. Although it has been demonstrated that Cdc14 released by the FEAR enhances MEN signaling by dephosphorylating Cdc15 and Mob1, which promotes their binding to the SPBs, it has also been shown that Tem1 activity is required to recruit Cdc15 to the SPBs. The fact that deletion of SPO12 prevented accumulation of Cdc15 on the SPBs in *kin4Δ* cells with mispositioned spindles (Figure 5E) would also be expected if Tem1 were inactive due to the inhibitory action of Bfa1. Thus, and again, this result does not allow excluding that what the FEAR mainly does is to facilitate Bfa1 inhibition, and consequently Tem1 activation. Along the same lines, the fact that a *cdc15-7A* allele promotes mitotic exit of cells with misaligned spindles both in the presence and the absence of the FEAR is completely expected, since the MEN is being activated downstream of Tem1. In fact, that fewer cells with a SPOC-deficient phenotype accumulated in *kar9Δ cdc15-7A spo12Δ* cells than in *kar9 cdc15-7A* (Figure 5F) is also in agreement with the possibility that FEAR is enabling the inhibitory action of Cdc5 on Bfa1, since lack of Spo12 would promote Tem1 inactivation, and therefore Cdc15 loading on the SPBs and MEN signaling would be less facilitated.

Sorry for the confusion. We agree with the reviewer that the binding of Cdc15 to SPBs is critical for mitotic exit, as it has been previously shown by the Amon and Schiebel labs. We do not exclude that FEAR facilitates Bfa1 inhibition, as our data shows that the SPOC proficiency of *kin4Δ spo12Δ* cells requires Bfa1-Bub2. However, we think that Bfa1-Bub2 regulation is not only under Kin4 control. It is clear from these experiments that *kin4Δ* and *bfa1Δ* cells behave differently in the context of *spo12Δ* (*kin4Δ spo12Δ* cells are SPOC proficient whereas *bfa1Δ spo12Δ* cells are not). As Bfa1 behaved similarly in both *kin4Δ spo12Δ* and *kin4Δ* cells, we considered that Bfa1 must be regulated by mechanisms that are independent of Kin4 and Cdc5. Our data suggest that FEAR facilitates mitotic exit in *kin4Δ* cells in the mother cell compartment at the level of Bfa1-Bub2 and Cdc15. To avoid confusion, we now expanded the discussion to add this aspect (Page 16).

4.-The authors suggest that Bfa1 phosphorylation by CDK is required to engage the SPOC arrest, and that lack of Spo12 in *kin4Δ* cells impedes the Cdc14-dependent dephosphorylation of Cdk-dependent sites on Bfa1, thereby rescuing SPOC function. To test their hypothesis, Caydasi et al. generated a mutant version of Bfa1 in which six Cdk consensus sites in the protein are mutated to alanine (Bfa1-6A). However, there are several issues regarding the Bfa1-6A protein. First of all, the authors show that Bfa1-6A cannot be phosphorylated in vitro by Clb2-Cdc28 (Figure 6A). However, Bfa1-6A seems to be phosphorylated in vivo to a similar level than wild type Bfa1 (Figure S6B). A more careful analysis of the phosphorylation status of the mutant Bfa1 proteins (similar to that shown in Figure 4A) would have been more appropriate to assess the in vivo consequences of the mutations.

We now show the running profile of Bfa1 and Bfa1 mutants on SDS-PAGE gels during the cell cycle progression (Supplementary Fig 9c), during cell cycle arrest upon MEN inactivation by Tem1-depletion (Fig. 6c and Supplementary Fig 9e) and upon checkpoint activation (Supplementary Fig. 9f). In cells with properly aligned spindles and inactive MEN (G1-arrest and release under conditions that allow Tem1-depletion), *Bfa1-6D* migrated slower than *BFA1* or *Bfa1-6A* (Supplementary Fig. 9e) and hypophosphorylated Bfa1 forms are more apparent in *bfa1-6A* in comparison to *BFA1-6D* or *BFA1* in *spo12Δ* cells (Supplementary Fig. 9e). In cells with misaligned spindles, we see no accumulation of hyperphosphorylated forms for Bfa1 or Bfa1-6A (Supplementary Fig. S9f). Nevertheless, unlike Cdc5, we could not assign a specific band to Cdk-phosphorylated Bfa1.

Furthermore, it should be also checked whether these mutations affect SPB localization of

Bfa1/Bub2, which is another important aspect of the regulation of the GAP.

The localisation of Bfa1 is not changed by the mutations. We now show in Supplementary Figure 9a that Bfa1 or Bfa1-6A/6D binds asymmetrically to SPBs (preferential binding to the budward SPB) in *kar9Δ* cells with normal aligned spindles but it binds symmetrically to SPBs (similar intensity at both SPBs) in cells with misaligned spindles. In cells with normal aligned spindles arrested in late anaphase, Bfa1 or Bfa1-6A/6D binds preferentially to the budward SPB (Supplementary Fig. 9b).

Finally, it is worth noting that phosphorylation of a protein by Cdc5 usually requires the priming phosphorylation by CDK of a Polo-binding site within the protein. Interestingly, the sequence of Bfa1 has a consensus Polo-binding site that includes amino acids S453 and S454, one of the amino acids mutated in Bfa1-6A. In fact, S454 is one of the four residues mutated in the Bfa14A mutant described by Kim et al., which cannot be phosphorylated by Cdc5 (Kim et al., 2012). Is the Bfa1-6A mutant still phosphorylated by Cdc5 in an *in vitro* assay similar to that shown in Figure 6A? If phosphorylation of the Bfa1-6A mutant by Cdc5 were affected, this would complicate the evaluation of the effects of the mutant on SPOC functionality.

This is a good point. We now show that Bfa1-6A is phosphorylated by Cdc5 *in vitro* similarly to Bfa1 (Supplementary Fig. 8). Furthermore, we found S454 to be phosphorylated *in vitro* by Cdk but not by Cdc5 (data not shown). However, we consider our negative MS-result for Cdc5 as not conclusive, as Kim et al could detect S454 phosphorylation of Bfa1 by Cdc5 *in vitro*. It is therefore possible, that Cdk and Cdc5 phosphorylates S454 *in vivo*. Nevertheless, if Cdc5 would phosphorylate S454 to inactivate the Bfa1-Bub2 complex, the S454A mutation in Bfa1 should assist Bfa1 to engage the SPOC, which is not the case. Furthermore, if Cdk would prime for Cdc5-phosphorylation of Bfa1, the Bfa1-6A mutant would be less phosphorylated by Cdc5 and therefore SPOC proficient (as Cdc5 phosphorylation of Bfa1 decreases Bfa1-Bub2 GAP activity). However, we found that Bfa1-6A is SPOC deficient. This argues against a priming function of Cdk. We now discuss this (Page 16).

Finally, some minor points are:

5.- Overexpression of SPO12 determines improper mitotic exit in cells with mispositioned spindles, while lack of Spo12 rescues the functionality of the SPOC in *kin4Δ* cells. However, this does not mean that the deficiency or the overexpression of Spo12 affect mitotic exit signaling in cells with mispositioned spindles merely due to opposite mechanisms. As such, and while overexpression of SPO12 suppresses the lethality caused by Bfa1 overexpression, lack of Spo12 does not rescue the SPOC deficiency in cells lacking BFA1. Therefore, and although the link between FEAR and the SPOC is later clearly established, the initial overexpression experiments do not really provide a basis for a possible role of Spo12 in the SPOC. On a different note, the fact that overexpression of Spo12 rescues the temperature sensitivity of many MEN mutants, but not of cells carrying *cdc14-ts* alleles, had been previously well established in the literature (e.g., see Jaspersen et al., 1998), so Figure S1B is not really required.

We agree with the reviewer. However, as we are comparing CEN and 2 μ -SPO12 constructs here, and our data provided comparison of several MEN mutants in the same experimental setup, we decided to leave this supplemental panel, although similar data were shown by Sue Jaspersen and colleagues in 1998.

6.- In Page 9, it is stated that "[...] Thus, lack of FEAR restores the SPB-bound Cdc15 and Mob1 levels in *kin4Δ* cells with misaligned spindles to prevent MEN activation". However, this sentence leads to confusion, since deletion of SPO12 in fact prevents Mob1-GFP and Cdc15-GFP loading on the SPBs in *kin4Δ* cells (Figures 5C-E). The sentence should be rephrase to avoid this confusion.

Thank you for pointing this out. We changed this passage.